# Detailed observations reveal the genesis and dynamics of destructive debris-flow surges

J. Aaron [1,2] ✉, J. Langham [3], R. Spielmann [1,2], J. Hirschberg [1,2], B. McArdell [2], S. Boss [2], C. G. Johnson [3] & J. M. N. T. Gray [3]

Debris flows are one of the most damaging natural hazards in mountainous terrain. Their dynamics are controlled by both surging behaviour and the influence of large boulders. However, a lack of high-resolution field measurements has limited our mechanistic understanding of these important processes. Here, we provide high-resolution in situ debris-flow surge measurements that demonstrate that surges are formed by the spontaneous growth of small surface instabilities into large waves, which amplify the destructiveness of the flow by increasing peak discharge. We use our field measurements to invert for the effective basal friction experienced by the flow, and support this reconstruction using numerical simulations that reproduce the formation and propagation of the surges. Detailed analysis of the inverted frictional data further shows that large boulders in the flow can influence local flow dynamics by increasing basal resistance, but this is not required to drive the surge wave instability. Our analysis provides new insights into debris-flow dynamics and can provide the foundation for improved hazard management of these damaging processes.

On the 11th August 2019, a mixture of soil, water and woody debris impacted the village of Chamoson (Valais, Switzerland) killing two people[1]. Previous debris flows in Chamosan have been documented to contain multiple surges, which amplify flow discharge and increase destructive potential. Approximately 3 years later, on June 5th, 2022, a debris flow occurred in a neighbouring catchment (Illgraben) that generated a series of nearly 70 surges. This event was documented by a monitoring system that recorded the surges with uniquely high spatial (~2 cm) and temporal (10 Hz) resolution. The resulting data are analysed herein.

Increased development pressures, as well as climate change, are changing debris-flow risk, which is highest on debris-flow fans[2]. The presence of surging behaviour in debris flows has been discussed by previous researchers[3–8], however the underlying physical mechanisms that lead to this important behaviour remain poorly understood. It has been suggested that surges are caused by longitudinal sorting[6,9], changes in bed topography[10], as well as the growth and coalescence of small disturbances in the flow[5,11–13]. Direct quantitative field evidence of these phenomena are rare in the literature[14], which strongly limits our ability to understand and predict debris-flow surges. Remarkably, this sparse documentation has included accounts of waves propagating through otherwise stationary debris[4,15].

The dynamics of debris flows are strongly coupled to the frictional resistance that they experience at their base. No consensus has yet emerged about what controls debris-flow friction. However, it has been suggested that it is affected by coarse-grained components within the flow[6,9,16], pore pressure effects[9,17–19], as well as flow depth and velocity[4,9,11,12,20–23]. Debris-flow surges can manifest as regular trains of quasi-steady travelling waves (surge waves) that have been hypothesised[13] to occur for the same essential reason as the phenomenon of roll waves found in shallow flows of laminar[24] and turbulent water[25,26], as well as dry granular media[20,21] and many other complex fluids[27–29]. However, obtaining compelling evidence in support of this view requires measurements of the spatiotemporal development of surges, together with a characterisation of the debris flow friction.

Roll waves are characterised by downstream propagating undulations of the flow surface led by a steep shock in which discharge is preferentially concentrated. They grow spontaneously from small perturbations due to an instability that is driven by gravity and the resultant frictional feedback of the flowing material[27]. A closely related class of pulses, termed erosion-deposition waves, can occur in materials featuring an effective yield stress, such as dry granular flows[12]. These propagate through recently deposited debris by mobilising a layer of recently deposited debris en masse at their fronts and redepositing it at their trailing edge. This mechanism is distinct from the typical view of debris-flow erosion, which relates to the mobilization of the static bed material along the path due to collisional stresses

[1]Geological Institute, ETH Zurich, Zurich, Switzerland. [2]Swiss Federal Institute for Forest Snow and Landscape Research (WSL), Birmensdorf, Switzerland. [3]Department of Mathematics and Manchester Centre for Nonlinear Dynamics, University of Manchester, Manchester, UK. ✉e-mail: jordan.aaron@eaps.ethz.ch

transmitted from the flowing debris to the bed material[30,31]. For flows whose frictional properties are well constrained, detailed predictions of the size, shape and speed of both wave types can be made[12,20,25]. By contrast, the difficulty of obtaining direct quantitative field measurements of full-scale debris flows has left the rheology of these flows underdetermined, thereby limiting our ability to understand their surge development and reliably model the associated hazards.

In the present work, we leverage a series of 3D laser (LiDAR) scanners, originally developed for autonomous driving, high-framerate video cameras, and machine-vision algorithms, which can image natural debris flows at over two orders of magnitude more detail than was previously possible[32–36]. We first present high-resolution measurements of the depth and velocity of the June 5th, 2022, debris flow, recorded at three separate observation stations throughout the duration of the event. Next, we estimate a continuous timeseries of flow discharge and use the theory of non-uniform unsteady shallow flow to infer the time-varying bulk basal friction experienced by the moving debris during surge-wave propagation. Finally, we use these measurements to specify both the input hydrograph and the friction law of a state-of-the-art numerical model. The resulting simulations validate the field data analysis and provide further insight into the formation and propagation mechanisms of the documented surges.

## Results and discussion
### Surge observations
We describe a debris-flow event that occurred on June 5th, 2022, in the Illgraben, Switzerland (Fig. 1). This event was recorded by five LiDAR scanners and six video cameras which are installed at three check dams (CD), as well as a force plate. These sensors are located at three observation stations on the Illgraben fan and integrated into the debris-flow monitoring network run by the Swiss Federal Institute for Forest, Snow and Landscape Research (WSL) (Fig. 1A). We divided the 3D LiDAR data at each station into a series of cross sections (Supplementary Figs. S1, S4 and S6), and used these sections to derive debris-flow properties, as described in the "methods" section. These properties include the surface velocity, the width- and depth-averaged material velocity (hereafter referred to as velocity), as well as the velocity of surges as measured by the propagation speed of wave crests (hereafter referred to as the wave velocity[37,38]), the section-averaged depth (estimated relative to the pre-event topography, and hereafter referred to as depth), the velocity of individual boulders and woody debris, as well as the spatial and temporal derivatives of depth and velocity. All three locations featured a bouldery front for this event (Supplementary Videos S1, S2 and S3), with front velocities decreasing from 5.5 m/s to 2.8 m/s as the flow travelled for about 1.7 km down the Illgraben fan between the stations Gazoduc and CD 29[39] (Fig. 2).

At the most upstream station (Gazoduc, Fig. 1A, B) we find that velocities are highly unsteady, with fluctuations up to 3 m/s over timescales of 10 s (Fig. 2A, B). We verified these velocity fluctuations by manually mapping the trajectories of 160 features (boulders and woody debris) in the LiDAR data (Fig. 2A). The minimum surface velocities correspond to times when large boulders are moving through the measured channel reach and the maximum velocities correspond to woody debris being transported on the flow surface, consistent with previous work[32]. Interestingly, the observed velocity fluctuations do not correspond to fluctuations in depth for the first 10 min (Fig. 2B, confirmed by analysis of the videos), indicating that they are not surge waves.

The measured debris-flow behaviour completely changes at the downstream measurement stations (CD 27 and CD 29), where the flow has transitioned into a series of waves (Supplementary Videos S4, S5), whose velocity (wave vel, green dots in Fig. 2C, D) substantially exceeds the measured material velocity (velocity, red line in Fig. 2, Supplementary Video S6). We measure 66 surges at CD 27 over a 30 min time period (2.2 surges/minute), and 42 surges at CD 29 over a 20 min time period (2.1 surges/minute) (after which the scanner malfunctioned). The similarity of surge frequency at these two measurement stations indicates that few of these surge waves coalesced over the 450 m in between the two monitoring

stations. Figure 2D also shows that there are periods where the debris-flow material becomes completely stationary (Supplementary Video S5) and is then subsequently remobilized by the passage of a wave. These phenomena are strikingly similar to erosion-deposition waves observed in dry granular flows[12] and only occurred at the most downstream station during this event (CD 29). Remarkably, compression of material at the front of the waves can be seen in the LiDAR data, suggesting the wave crests extend below the surface of the flow (Supplementary Videos S7 and S8). Furthermore, while the flow depth increases between CD 27 and CD 29, velocities decrease between these two measurement locations (Fig. 2C, D), consistent with the reduction in front velocity measured between the two stations. These changes in flow variables occur despite relatively constant channel widths and cross-sectional areas between the two stations (Supplementary Figs. S4 and S6), although there is a gradual change in slope angle, as investigated below.

### Discharge and basal friction
These measurements were used to explore the influence of surges on debris-flow discharge, as well as to understand the mechanisms governing surge formation and movement. We used the velocity and depth timeseries to estimate the discharge of material through a representative cross section, as well as to invert a pair of depth-averaged shallow flow equations (described in the methods section) to estimate the effective bulk friction acting at the base of the debris flow. We validated the inverted friction by comparing the resulting coefficient to corresponding measurements from a large force plate installed at CD 29 (Supplementary Fig. S10).

The evolution of discharge, which controls debris-flow destructiveness, shows pronounced spatial and temporal variations (Fig. 3). At the most upstream measurement station (Fig. 3A), peak discharge occurs a few seconds behind the arrival of the front and falls off throughout the event. In strong contrast, peak discharge occurs well behind the front at the more downstream locations (Fig. 3B, C), demonstrating that the surge waves cause peak discharge values that substantially exceed the front discharge.

The friction coefficients inverted at the three stations show pronounced differences (Fig. 3). At Gazoduc, the inverted basal friction coefficient shows that the flow is unsteady, with basal resistance fluctuations correlated to the presence of large boulders (Fig. 2A). This is consistent with previous hypotheses[6], and represents a clear demonstration that large particles within a debris flow can locally increase flow resistance, both at and well after the passage of the bouldery flow front. At CD 27, there are more high-frequency variations than at CD 29, which also correspond to the presence of boulders in the measured cross sections. We observe fewer boulders at CD 29 than at CD 27, which may be due to boulder deposition between the two stations or because boulders become obscured when the flow depth rises. At both CD 27 and CD 29, friction decreases at peaks in the discharge timeseries (Fig. 3), which is consistent with shallow-flow analyses of roll waves[37].

### Surge formation and propagation
To understand the mechanisms governing surge dynamics, we performed numerical simulations of the same flow equations used in the friction coefficient inversion procedure, with the Kestrel open-source shallow flow software[40] (Fig. 2E-G). Our approach here is to provide insights into the formation and dynamics of the measured surges while including the minimal amount of complexity, rather than to extensively calibrate a hazard model. We therefore used an existing granular friction law[12,41,42] adjusted to quantitatively approximate the inferred field values of basal friction across the physical regimes recorded during the event (Fig. 4), and use a simplified 1D topography that smoothly transitions between the measured slope angles at Gazoduc and CD 27 (4.5°) and CD 29 (3.5°). Details of the model friction law and the process of parameter selection are given in the "Methods" section. As discussed later, the effects of large boulders on friction are not explicitly included in our simulations.

Waves spontaneously arise in the simulation from an underlying roll wave instability[23,41] that is present for all flow conditions with the model

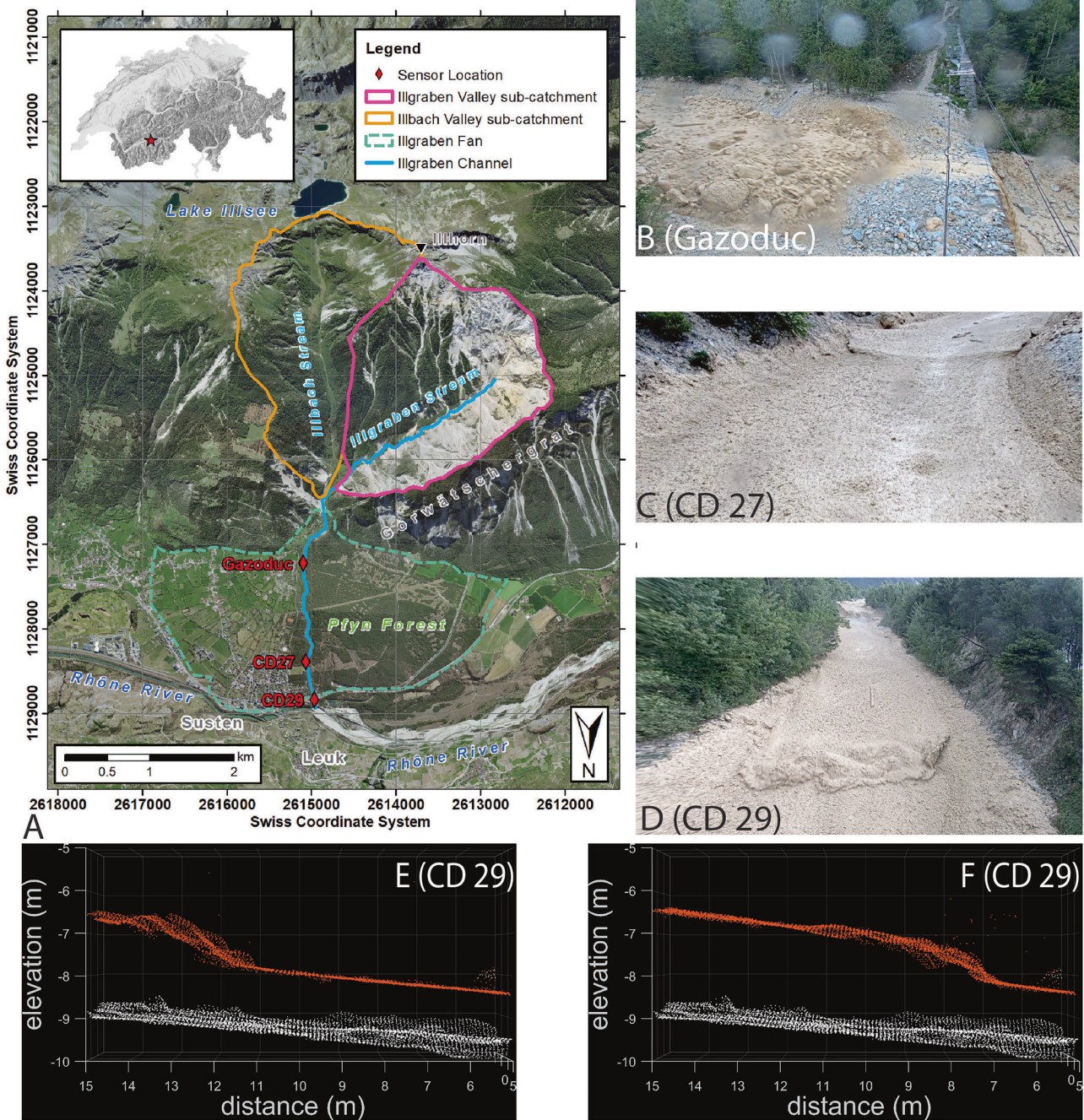

**Fig. 1 | Overview of the study site and monitoring setup. A** Overview map of the Illgraben, showing the location of the three observation stations. The distance between the three stations, from Gazoduc downwards, is 1250 m and 450 m. Image: © swisstopo. **B** Bouldery front arrival at the most upstream station, referred to herein as Gazoduc. **C** Surge wave at CD 27. **D** Surge wave propagating through initially stationary material at CD 29. **E, F** Surge wave shown on image (**D**) at two different times in the LiDAR data, with the white dots showing the pre-event scan of the channel bottom, and the orange dots showing the instantaneous flow surface. The frames are 0.7 s apart.

friction law. The simulations are effective at reproducing the observed depths and velocities at each measurement station, despite not being explicitly calibrated against these measurements (Fig. 2), and trace out essentially the same regions of depth-Froude number space (black lines on Fig. 4) as the corresponding values from the directly measured surges (coloured points on Fig. 4). The data separation in Fig. 4 demonstrates that the relatively modest change in slope between the two stations (1°) influences the transition of the observed surges between two regimes, of faster, thinner roll waves at CD 27 and slower, thicker erosion-deposition waves at CD 29 (Figs. 2 and 4). For comparison, a simulation was performed using the same conditions, on a constant 4.5° slope. This undergoes the same

instability development, ultimately leading to erosion-deposition waves, but lacks the clear separation between the CD 27 and CD 29 data observed in the field (see Supplementary Fig. S 12).

The simulation and field data mutually indicate that the large amplitude surges at CD 27 and CD 29 are caused by the growth and coalescence of small instabilities as the flow moves down the fan. This process is clarified in Fig. 5, which shows a sequence of snapshots of the simulated flow depth along the section between Gazoduc and CD 29, at 150 s time intervals. The initial thick (1.5 m) inflow at Gazoduc is approximately uniform upstream of the front (Fig. 5A). As the flow propagates towards CD 27, small disturbances become visible (Fig. 5B), which grow into substantial waves,

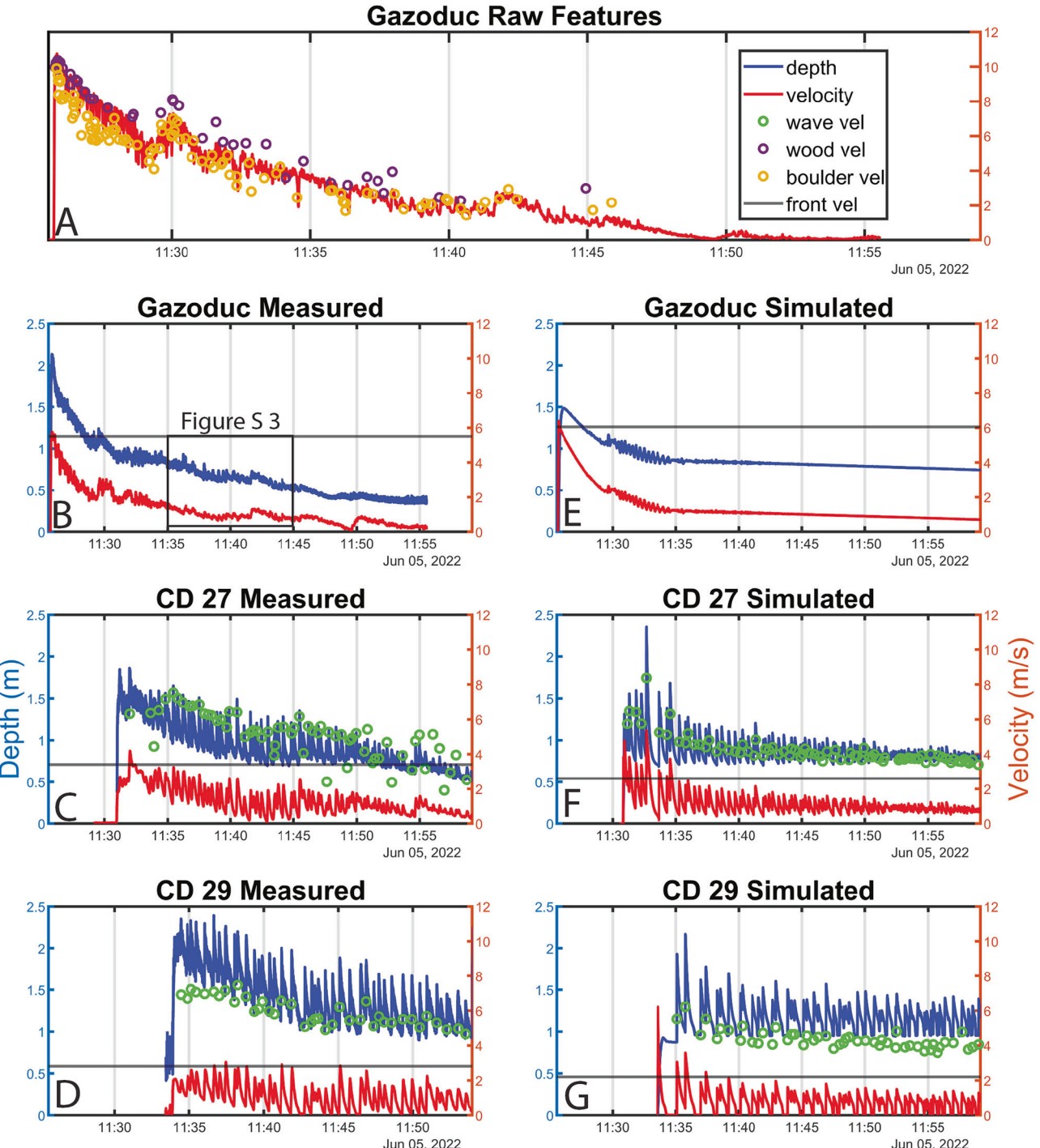

**Fig. 2 | Measured and modelled surface velocities and depths. A** Surface and feature[39] velocities measured in the channel center at Gazoduc. These velocities are not width- and depth-averaged. **B–D** Velocity and depth measured at a representative cross section at each of the measurement stations (see Supplementary Figs. S1, S4 and S6). **E–G** Velocity and depth simulated using a single-phase shallow flow model.

reaching amplitudes of half a metre or more by the time they reach CD 27 (Fig. 5C). Alongside this, the flow thins due to the waning of the upstream flux, and briefly becomes substantially more unstable, as evidenced by the appearance of larger waves near Gazoduc (Fig. 5C, D), and corresponding to field observations (Fig. S3 in the supplement). Furthermore, the mature roll wave velocities are much faster than the material and front velocities, in line with the field observations (Fig. 2). For example, the wave highlighted in

solid red gains roughly 400 m relative to the front position in the 90 s between Figs. 5C, D. Just prior to the snapshot in Fig. 5E, it catches the front (which decelerates on the shallower gradients at CD 29) leaving a large deposit (blue shaded region on Fig. 5E). The later panels (Fig. 5D-F) contain four static deposits, indicated with blue shading. An erosion-deposition wave, separated by two such regions, can be seen travelling between CD 27 and CD 29 (Fig. 5F). The full simulation continues for ~15 min after the final

**Fig. 3 | Measured discharge and inverted basal friction.** Discharge (solid blue) and inverted friction coefficient (solid red) at the three measurement stations (**A** to **C**). The horizontal black lines mark the friction values equal to the tangent of the local slope angle, which occur under steady uniform flow conditions (see eq. [S1] in the supplement).

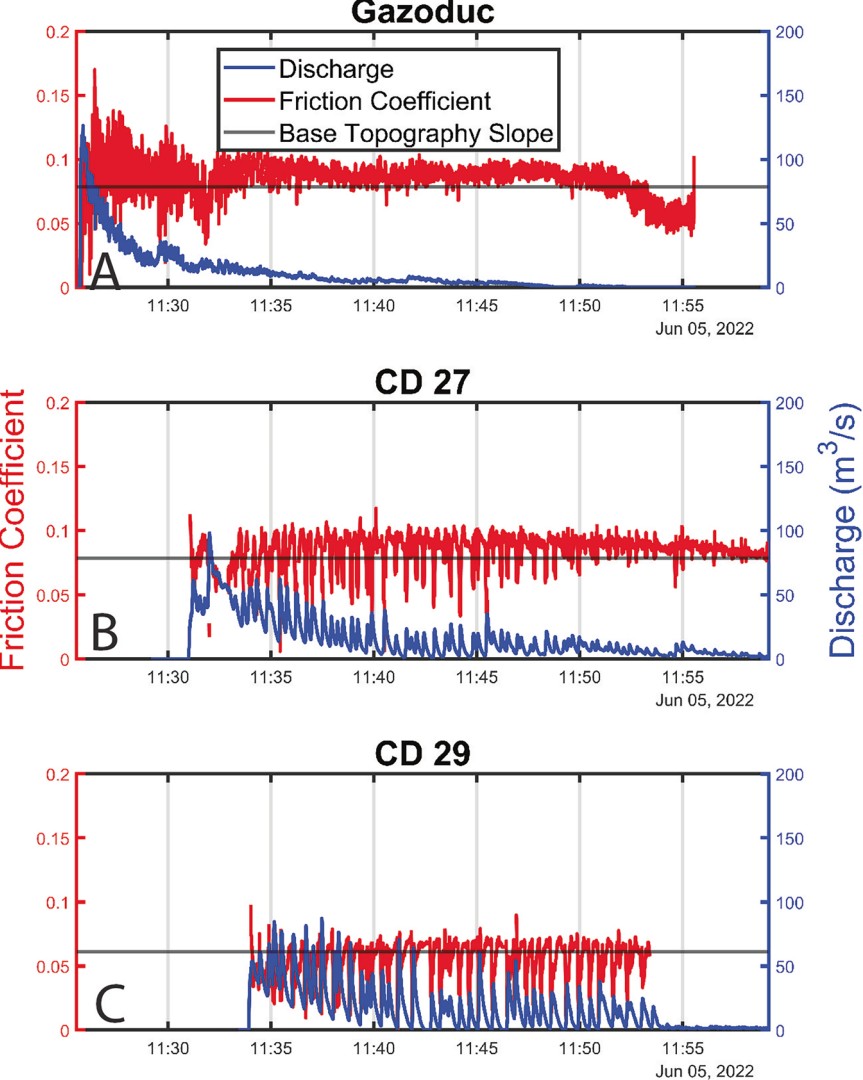

**Fig. 4 | Relationship between friction coefficient, depth and Froude number for CD 27 and CD 29.** The black lines show the friction coefficient for each of the simulated waves, which demonstrates that the simulations trace out the same region of the parameter space as the measured data.

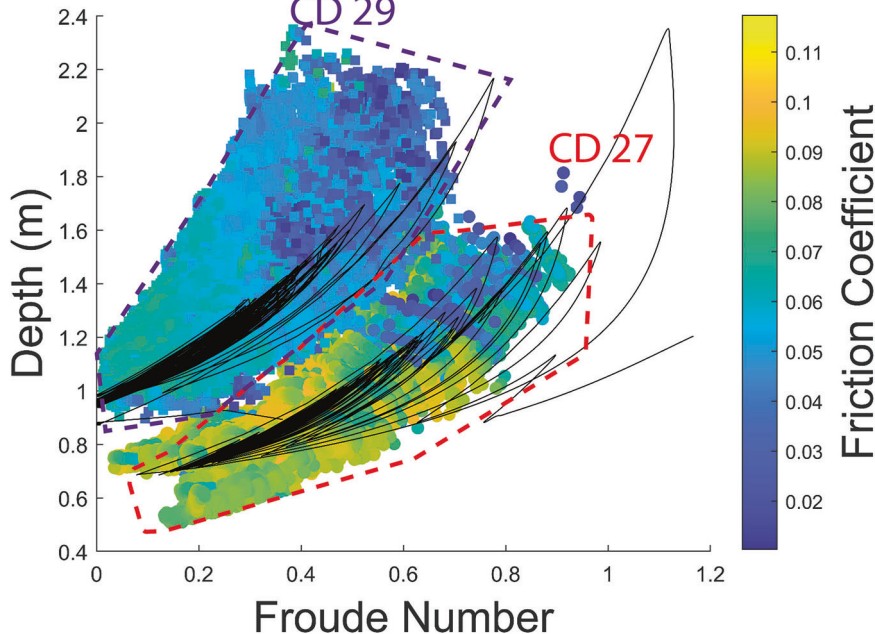

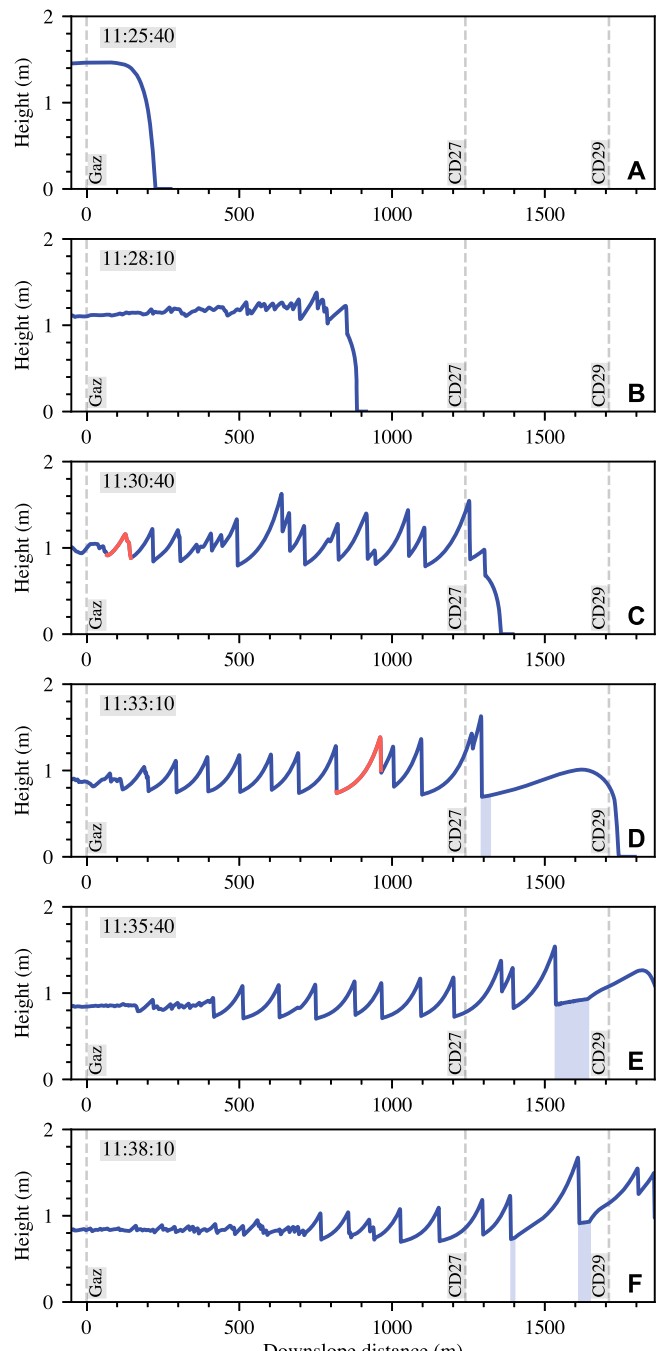

**Fig. 5 | Numerical modelling results.** Snapshots of simulated flow depth versus the downslope distance from Gazoduc, separated by 150 s intervals (**A**–**F**). An individual waveform is highlighted with a solid red line in (**C**, **D**) to visualise its propagation and development relative to the rest of the flow. Grey dashed lines indicate the locations of the three measurement stations and regions of static deposit are shaded light blue.

snapshot and may be viewed as an animation provided in the Supplementary Material (Supplementary Video S9).

### Implication for debris-flow mechanics and hazard

Our measurements of in situ debris-flow surges are among the highest combined spatial and temporal resolution ever collected. Compared to previous studies[5,13,14,43], they allow us to understand the length scales over which surge waves initiate, their control on debris-flow hazard, as well as the basal resistance experienced by the flow during surge wave generation and propagation. These surge waves magnify peak discharge by two to three times (relative to the front discharge), greatly increasing flow depth and velocity (Fig. 3). Accounting for these surge waves is thus critical in debris-flow hazard analysis, as our measurements show that their presence increases the potential destructiveness of the flow as it travels down the fan.

The numerical modelling results demonstrate that the formation and coarsening of surge waves can be quantitatively captured using a single-phase 1D depth-averaged model, so long as the friction law and channel slope angles are appropriately selected (Figs. 2 and 4). Given the shallow gradients of the Illgraben fan, a change in slope angle of just 1° leads to a measurable change in flow depth and velocity (Figs. 2 and 4) that is commensurate with the field data, thereby implying that debris-flow dynamics can be sensitive to seemingly gradual topographic variations[44]. Our simulations cannot capture all the details of the flow, in particular the front shape and behaviour at the tail. This is likely due to the fact that various time-varying properties of the liquefied slurry[17,45–47] are not accounted for. For example, the field measurements at Gazoduc and CD 27 demonstrate that large boulders locally increase the basal resistance. This leads to the presence of relatively greater frictional variations over the length of the bouldery front, and unsteady flow behind the front, consistent with observations made by other researchers[6,9,16,48].

Additionally, pore pressure generation and dissipation is known to play a role in governing debris-flow friction[17,47,49–52]. Our results show that low bulk basal friction angles, on the order of 3–6° occur throughout the flow (Fig. 3C, D), which implies substantial pore pressures developing in the flowing debris. Nevertheless, we are able to simulate many bulk characteristics of the flow without explicitly accounting for spatial and temporal variations in pore pressure[49,50,53] (Fig. 2). Consequently, we conclude that our newly developed friction inversion procedure is able to aggregate complex multi-phase debris flow physics into an apparent friction coefficient that can be implemented in single-phase depth-averaged models. While we have only applied the new methodology to a single event, our analysis provides the foundation for a more generalized, practical model that avoids the added complications and computational cost inherent in multi-phase and non-depth-averaged systems. This should become possible once the analysis procedure has been applied to a large number of debris-flow events.

A key implication of our findings is that debris flow surges can emerge spontaneously as the result of a linear instability, and that the resulting surge waves can manifest as either roll waves or erosion-deposition waves. This mechanism has been hypothesized by previous researchers[11–13], however direct high-resolution measurements with which to test these hypotheses have been lacking. Furthermore, no consensus on the correct rheological description for modelling debris flows exists, which is critical for effectively capturing roll wave instabilities in numerical models. The close agreement between our data and the numerical simulations both supports the existence of the underlying instability in the field and provides empirical support for our tested friction law. Interestingly, analysis of this friction law, with the parameters in the presented simulation, implies that the flow is unstable regardless of Froude number, and that this condition is required for surge waves to form, regardless of compositional variations (as elaborated on in the Methods). However, the (linear) spatial growth rate of the resulting surges is highly dependent on this quantity. In particular, the dependence is non-monotonic, reaching a maximum at intermediate values and ultimately decaying to zero as Froude numbers become very high[54]. Our data and simulations support this interpretation, as shown in Fig. 5, as well as Fig. 2B, which demonstrates that no surge waves are present at Gazoduc until the flow depth and velocity have waned, and then small surges start to appear in the flow after about 10 min (see also Supplementary Fig. 3). Similarly, Fig. 5C, D shows a period of increased instability as the source flux at Gazoduc wanes in the simulation. This suggests that the high discharge surges documented in the present event become most pronounced at intermediate Froude numbers, and would not have arisen in far faster, or far slower flows of the same composition. Continued application of theoretical insights of this kind, which can draw upon an extensive history of roll wave analyses in other systems, may soon help to disentangle open questions in

debris flow science, such as why some events produce substantial surge trains, while others do not.

In summary, our results provide a new perspective on debris-flow motion. The field measurements show that field-scale debris flows are spatio-temporally complex and can transition between different flow and surge regimes as they move down a fan. These complex dynamics directly control debris-flow destructiveness and therefore must be accounted for when managing debris-flow hazards. Remarkably, we find that the measured phenomenological complexity is governed by two main mechanisms: increased flow resistance due to the presence of large boulders, which cause highly unsteady velocities, and the development of surface instabilities into damaging waves. While there is likely some coupling between these two mechanisms, with large boulders generating surface disturbances that grow into surge waves, we conclude that the linear instability can occur without explicitly accounting for the effects of large boulders. Furthermore, we show that the instability can be captured using a single-phase numerical model which quantitatively reproduces spatio-temporal variations in surge velocities, depths and types, and reveals that debris-flow dynamics can be acutely sensitive to the channel slope angle. The striking effectiveness of our relatively simple modelling approach suggests that suitably parameterized, computationally inexpensive models could provide a promising path towards tools for markedly improved debris-flow hazard assessment. Incorporating the effects of composition variation and, in particular, the influence of large particles[55,56], could lead to yet greater fidelity.

## Methods
### Measurement of velocity and section-averaged depth
**Material and wave velocity.** We derived a dense surface velocity field by using the particle image velocimetry (PIV) based hillshade methods[32,33], using open source packages for PIV[57] and hillshade generation[58]. This method is sensitive to the surface features and texture of the flow, and therefore represents a material velocity (which is contrasted to the wave velocity measured by manually tracking the wave crests of surge waves). As shown on Figs. S1, S4, and S6, the application of these methods resulted in a dense set of surface velocity vectors (one vector approximately every $1 \times 1$ m) in the measured channel reach at each of the monitoring stations. As described below, we subsequently width- and depth-averaged these velocities, and used those for further analysis.

We validated our surface velocity algorithm by comparing PIV-derived velocities with those corresponding to features manually mapped in hillshade projections of the LiDAR point clouds. These features were manually mapped by drawing bounding boxes around identified features in multiple frames, and velocities of the features were derived by calculating the distance between the bounding box centers in subsequent frames, and dividing by the timespan between them. As shown in Fig. 2A, the correspondence is quite high, with maximum values corresponding to woody debris and minimum values to large boulders.

We used a similar procedure as that used for feature mapping to manually map the crests of surge waves as they move through the measured channel reach (referred to as wave velocity in the text). As each wave is mapped multiple times per station, we measure multiple instantaneous velocities, which we subsequently average to derive a mean velocity for each surge wave along the monitored channel segment of each measurement station.

**Average depth.** We derived estimates of section-averaged depth (defined to be the mean flow depth across the wetted cross-section perpendicular to the bulk flow direction) by assuming a base topography in the channel corresponding to the pre-event topography. This is likely a reasonable estimate at our study site, as the presence of check dams stabilizes the bed near our measurement locations. We estimated the cross-sectional width by measuring the distance between the instantaneous flow surface and the intersection with the pre-event topography, and estimated the cross-sectional area by estimating the area between the pre-event topography and the instantaneous flow surface

(Figs. S1, S4, S6). We then computed the average depth by dividing the cross-sectional area by the flow width. The depths across the selected representative sections (see below) can be seen on Supplementary Figs. S1, S4 and S6.

**Section-derived quantities.** We subdivided each measurement station into 50 individual cross sections (Supplementary Figs. S1, S4 and S6). For each cross section, we derived a width-averaged surface velocity ($u_s$) as well as the average depth (as described above). We then selected representative cross sections for each station (red section on Supplementary Figs. S1, S4 and S6), and used these for subsequent analysis. To estimate $u_s$, we first fitted an interpolant to the instantaneous velocity field (Figs. S1A, S4A and S6 A), and then evaluated this interpolant along each analysed cross-section (Figs. S1B, S4 B and S6 B). We then calculated the average of $n$ along-section velocities ($u_i$) per cross section, with each $u_i$ spaced 0.1 m apart (Figs. S1C, S4C and S6C), to obtain the width-averaged surface velocity ($u_s$):

$$u_s = \sum_{i=1}^{n} \frac{1}{n} u_i. \tag{1}$$

Next, we estimated the depth-average of the width-averaged velocities for each section

$$\bar{u} = \alpha u_s, \tag{2}$$

where $\alpha$ is the surface velocity reduction factor (0.7) estimated based on the relative velocities of woody debris and boulders[32,33], and consistent with other studies[14,47]. It should be noted that this coefficient likely varies throughout the debris-flow[32,47], and we have used a single constant value for simplicity. We then estimated discharge by multiplying the resulting depth-averaged velocity by the cross-sectional area to obtain discharge.

Froude numbers ($Fr$) were also estimated for each cross section, according to the definition:

$$Fr = \frac{\bar{u}}{\sqrt{h g \cos \theta}}, \tag{3}$$

where $h$ is the section-averaged depth (described above), $g$ is the gravitational constant and $\theta$ is the channel slope angle.

### Friction slope inversion
We invert a shallow flow model in order to estimate the basal friction acting on the debris flow. A description of the variables used is shown in Fig. S8. The depth-averaged mass and momentum equations are

$$\frac{\partial h}{\partial t} + \frac{\partial}{\partial x}(h\bar{u}) = 0, \tag{4}$$

$$\frac{\partial}{\partial t}(h\bar{u}) + \frac{\partial}{\partial x}\left(h\bar{u}^2 + \frac{h^2 g \cos \theta}{2}\right) = hg \sin \theta - \mu_b hg \cos \theta, \tag{5}$$

where $\bar{u}$ is the depth-averaged velocity, $x$ the bed-parallel downstream coordinate, and $\mu_b$ is the basal friction coefficient. Assuming that the downslope curvature is negligible, Eqs. [4] and [5] may be combined to give

$$\mu_b = \tan \theta - \frac{\partial h}{\partial x} - \frac{1}{g \cos \theta}\frac{\partial \bar{u}}{\partial t} - \frac{\bar{u}}{g \cos \theta}\frac{\partial \bar{u}}{\partial x}, \tag{6}$$

where

$$\frac{\partial h}{\partial x} = \cos \theta \left(\frac{\partial y}{\partial x} + \sin \theta\right), \tag{7}$$

with $y$ denoting the elevation of the top surface.

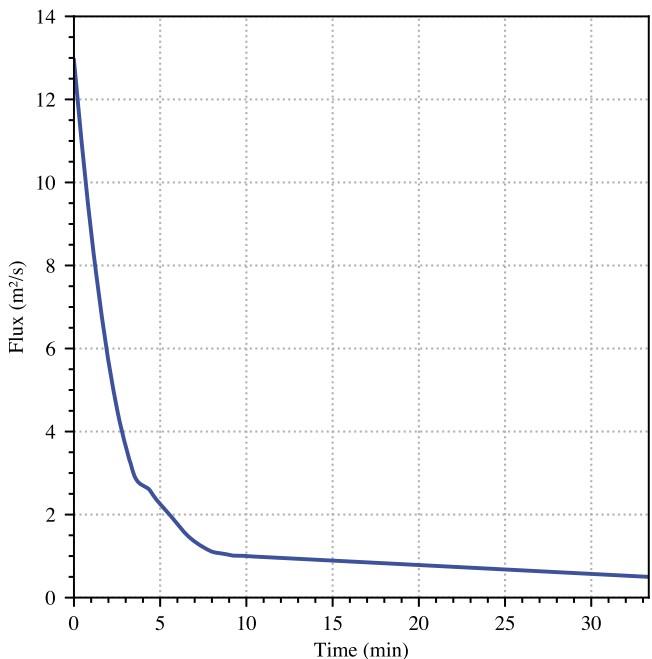

**Fig. 6 | Source condition used in the numerical simulations.** This is based on a scaled and smoothed version of the data provided in Fig. 2.

**Table 1 | Friction model parameters used in the numerical simulations presented in Figs. 2, 4, and 5**

| $\mu_1$ | $\mu_2$ | $\mu_3$ | $\beta$ | $\beta_\star$ | $\kappa$ | $L$ | $\Gamma$ |
|---|---|---|---|---|---|---|---|
| 0.00015 | 0.14 | 0.02 | 1 | 0.2 | 0.1 | 0.9 m | 0.8 |

We compute these formulae for the field data by evaluating all spatial and temporal derivatives based on width-averaged quantities, derived for each section as detailed above. It should be noted that estimating derivatives from the field data results in some high-frequency noise, which is apparent in the results of the inversion procedure.

We verify this inversion procedure by comparing the values derived from our equation to the ratio of shear to normal stress measured at the Illgraben force plate. The results of this are shown in Supplementary Figs. S9 and S10. It should be noted that the force plate is located at the brink of a free overfall, so we used the LiDAR measurements to correct for this effect by subtracting the inclination of the flowing surface from the measured shear/normal force values.

### Numerical simulations

We performed a simulation of the event, using the shallow flow equations [4] and [5] using the Kestrel software[40]. The underlying numerical solver is a well-balanced central-upwind finite volume scheme[59], with a constant grid spacing set to 0.5 m and the time step set adaptively to ensure that it does not exceed a Courant–Friedrichs–Lewy number of 0.5. To allow the slope angle to vary between two values between CD 27 and CD 29, a simple topography function was employed that smoothly connects between two slopes with angles $\theta = \theta_1$ and $\theta_2$ either side of $x = 0$ m. Specifically, we set

$$\theta(x) = \arctan\left\{\frac{1}{2}(\tan\theta_1 + \tan\theta_2) - \frac{1}{2}(\tan\theta_1 - \tan\theta_2)\tanh\left(\frac{x}{\lambda}\right)\right\},$$

(8)

where $\theta_1 = 4.5°$, $\theta_2 = 3.5°$ and $\lambda = 200$ m is the length scale over which the transition between the two angles occurs.

**Source condition.** Flow in the simulations is fed from a 2000 s (33.3 min) long time-varying source condition, applied as a forcing term added to the right-hand side of Eq. [4], within a 10 m radius centred around $x = -1975$ m. A suitable runout distance is required for the flow to adjust to the forces impinging on it, so we record data corresponding to the Gazoduc measurement station 500 m downstream at $x = -1475$ m. For CD 27 and CD 29, we record data at $x = -235$ m and $x = 235$ m. The separations of these locations match the distances between the stations in the field. The initial 10 min of the source function was constructed to approximate the field hydrograph by smoothing the product of the hydraulic depth and velocity measurements at Gazoduc in Fig. 2 and rescaling in time by a factor of 0.6, while keeping its area constant by rescaling its magnitude. The rescaling accounts for the deceleration of the resulting unsteady pulse between the source location and Gazoduc, while maintaining a consistent volumetric release. Finally, for the remaining simulation time, a linearly decreasing function, with a gradient chosen to halve the flux over the remaining time, was adjoined in a piecewise continuous manner. This procedure was designed to approximate the measured Gazoduc pulse, while allowing for a release that extends for the duration of the flow. A plot of the source flux is given in Fig. 6. The first 20 min of volume release recorded in the simulation is within 6% of the equivalent value measured at Gazoduc in the field.

**Friction Model.** The code was adapted for this paper to use the following non-monotonic friction model

$$\mu_b(Fr, h) = \begin{cases} \min\left(\mu_{start}(h), \left|\tan\theta - \frac{\partial h}{\partial x}\right|\right), & Fr = 0, \\ \left(\frac{Fr}{\beta_\star}\right)^\kappa \left(\mu_1 + \frac{\mu_2 - \mu_1}{1 + \frac{h\beta}{L(\beta_\star + \Gamma)}} - \mu_{start}(h)\right) + \mu_{start}(h), & 0 < Fr \leq \beta_\star, \\ \mu_1 + \frac{\mu_2 - \mu_1}{1 + \frac{h\beta}{L(Fr + \Gamma)}}, & Fr > \beta_\star, \end{cases}$$

(9)

where $\mu_{start}(h) = \mu_3 + (\mu_2 - \mu_1)/(1 + h/L)$ and $\mu_1, \mu_2, \mu_3, \beta, \beta_\star, \kappa, L$ and $\Gamma$ are parameter values to be discussed shortly.

This piecewise continuous function[42], was constructed to extend an established empirical law for dry granular flows (the $Fr > \beta_\star$ case[41]) for situations close to $(0 < Fr \leq \beta_\star)$, or at arrest $(Fr = 0)$, in a manner consistent with observations of laboratory flows at low $Fr$[12,42,60]. In particular, equations [4] and [5], paired with the friction model in equation [9] support the maturation of roll waves into erosion-deposition waves similar to those observed at Illgraben. The parameters used for the simulation in the main text were chosen to approximate the friction values reconstructed from the field data (Figs. 3, 4) and are given in Table 1. They are necessarily quite different from values deduced from experiments in past studies, which have typically been fitted to flows of sub-millimetre diameter monodisperse glass beads or sand particles. We provide a more detailed explanation of the parameter selection in the supplementary information.

Finally, we note that for roll waves to exist at all, the flow must be linearly unstable. Linear instability in the dynamic friction regime occurs[54] when

$$Fr_0 > \frac{2(1 - \Gamma)}{3},$$

where $Fr_0$ is the Froude number of the corresponding uniform flow that generated the instability[37].

If the flow is only unstable above some critical Froude number $Fr_c$ that is strictly greater than $\beta_\star$, then waves may either fail to develop entirely, or (depending on the magnitude of $Fr_c$) become stabilised as the upstream flux wanes. This latter case is demonstrated in the flow depicted in Fig. 7, for which $\Gamma = 0.6$, ultimately causing the flow to stabilise when $Fr_0$ drops below $0.2\dot{6}$. Consequently, the flow is only unstable during the initial stages where

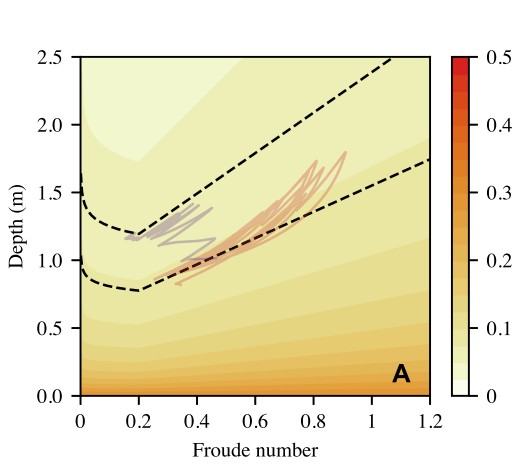
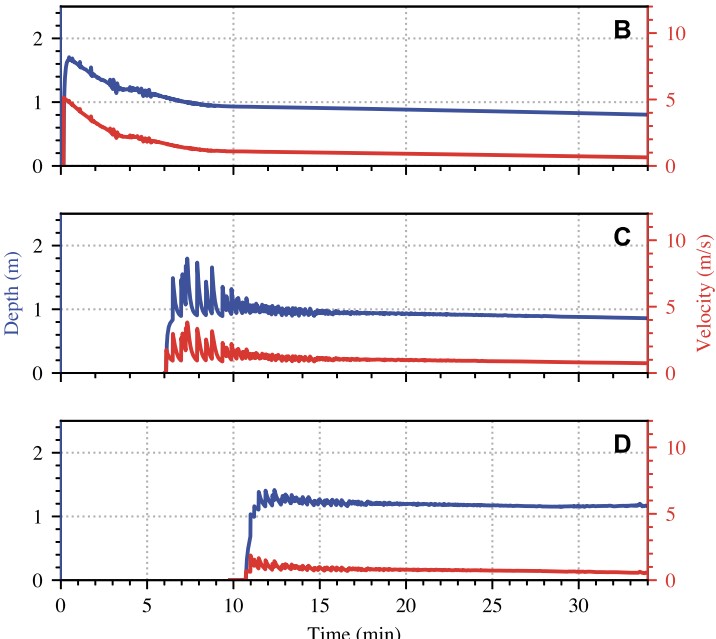

**Fig. 7 | A simulation using friction parameters that are not unstable for all *Fr*.**
**A** Contours of friction, overlaid with the simulation data at CD 27 (solid red curves) and CD 29 (solid purple curves), as well as the uniform steady balances (Eq. [S1]) with friction set by Eq. [9] at the two stations (black dashed). Depth and velocity at **B** Gazoduc, **C** CD 27, and **D** CD 29. The parameters are: $\mu_1 = 0.02, \mu_2 = 0.3, \beta = 3.5, \kappa = 0.03, \Gamma = 0.6$.

the input flux is high. Few waves survive at CD 29 and those that do, decay to leave a uniform steady flow, contradicting the flow observations at Illgraben. Therefore, since the $Fr < \beta_\star$ branch of the friction law is also unstable, this implies that we must select $\Gamma$ so that $\Gamma > 1 - 3\beta_\star/2$.

## Data availability
All data used in the publication, as well as the original data for the Figures, can be found at the following link: https://doi.org/10.3929/ethz-b-000736836.

## Code availability
The field data were processed using the following open-source software: • PIV-Lab https://www.mathworks.com/matlabcentral/fileexchange/27659-pivlab-particle-image-velocimetry-piv-tool-with-gui[57]. Hillshade Function: https://www.mathworks.com/matlabcentral/fileexchange/32088-esri-hillshade-algorithm[58]. The simulations presented in this paper were performed using a development branch of the Kestrel shallow flow solver that implements the friction law in Eq. [9]. Commit 925837d of the github repository was used throughout, available at github.com/jakelangham/kestrel. Note that earlier or later iterations of the codebase may not fully reproduce our results. Additionally, trivial modifications were made to output time-dependent point data at the measurement station locations. The input file for the main simulation run is available in the supplementary materials.

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

## Acknowledgements

Funding for this work was provided in part by an SNSF grant to J.A. (grant number 193081), as well as by NERC grants (NE/X00029X/1 and NE/X013936/1) given to J.M.N.T.G. and C.G.J. J.M.N.T.G. was also supported by a Royal Society Wolfson Research Merit Award (WM150058) and an EPSRC Established Career Fellowship (EP/M022447/1). The work was further funded by internal funding from the Swiss Federal Institute for Forest,

Snow and Landscape Research to B.M. Comments from three anonymous reviewers substantially improved the manuscript. We are also grateful to A. Badoux and C. Graf for their support with the field monitoring.

## Author contributions

J.A. conceived of and supervised the study, acquired funding, co-led the design, maintenance and installation of the monitoring system, processed the field data, contributed to the theoretical and numerical modelling and co-led the writing of the manuscript. J.L. led the implementation of the numerical model, performed the simulations, assisted with field data processing and theoretical modelling, and co-led the writing of the manuscript. R.S. assisted with field data collection and processing, contributed to the theoretical and numerical modelling, and writing of the manuscript. J.H. assisted with writing, as well as field data collection and processing. B.M. assisted with field data collection and contributed to writing the manuscript. S.B. co-led the design and maintenance of the field installation system and contributed to writing the manuscript. C.G.J. acquired funding for this work, contributed to the field data analysis, the theoretical and numerical modelling, and the writing of the manuscript. J.M.N.T.G. acquired funding for this work, contributed to the field data analysis, the theoretical and numerical modelling, and the writing of the manuscript.

## Competing interests

The authors declare no competing interests.
