## [Transparent Peer Review file · Communications Earth & Environment]

Detailed observations reveal the genesis and dynamics of destructive debris-flow surges

Corresponding Author: Professor Jordan Aaron

Version 0:

Decision Letter:

Dear Professor Aaron,

Your manuscript titled "Detailed observations reveal the genesis and dynamics of destructive debris-flow surges" has now been seen by 3 reviewers, whose comments are appended below. You will see that they find your work of some potential interest. However, they have raised quite substantial concerns that must be addressed. In light of these comments, we cannot accept the manuscript for publication, but would be interested in considering a revised version that fully addresses these serious concerns. Specifically, a revised manuscript must:

1. Fully address the limitations of the single-phase model in capturing the effects of boulders and multi-phase interactions, with a clear justification for its use and an explanation of how it can be applied for future predictions.
2. Clarify how field data were used to calibrate the model, provide a transparent explanation of the parameter selection process, and include a detailed error analysis to demonstrate how the calibration minimizes discrepancies between observed and simulated data.
3. Provide a more detailed explanation of "roll waves" and "erosion-deposition waves", particularly in relation to key flow properties such as flow depth and basal friction, to enhance the understanding of their influence on surge dynamics.

We hope you will find the reviewers' comments useful as you decide how to proceed. Should additional work allow you to address these criticisms, we would be happy to look at a substantially revised manuscript. If you choose to take up this option, please either highlight all changes in the manuscript text file, or provide a list of the changes to the manuscript with your responses to the reviewers.

When resubmitting, please provide a point-by-point response to the reviewers' comments. Please submit your responses as a separate file, distinct from your cover letter where you can add responses to the Editors' comments that you do not want to be made available to the reviewers. Word files are preferred. We recommend that any figures, tables or graphs that are included in the response to reviewers are also included in the main article or Supplementary Information.

If the revision process takes significantly longer than three months, we will be happy to reconsider your paper at a later date, as long as nothing similar has been accepted for publication at Communications Earth & Environment or published elsewhere in the meantime.

Please use the following link to submit your revised manuscript, point-by-point response to the reviewers' comments with a list of your changes to the manuscript text (which should be in a separate document to any cover letter), a tracked-changes version of the manuscript (as a PDF file) and any completed checklist:

Link Redacted

Please do not hesitate to contact us if you have any questions or would like to discuss the required revisions further. Thank you for the opportunity to review your work.

Best regards,

Alireza Bahadori, PhD
Associate Editor
Communications Earth & Environment
Consulting Editor
Communications Sustainability

EDITORIAL POLICIES AND FORMAT

If you decide to resubmit your paper, please ensure that your manuscript complies with our editorial policies and complete and upload the checklist below as a Related Manuscript file type with the revised article:

Editorial Policy Policy requirements
(Download the link to your computer as a PDF.)

- Behavioural and social science
- Ecological, evolutionary & environmental sciences
- Life sciences

<https://www.nature.com/documents/nr-reporting-summary.zip>

For your information, you can find some guidance regarding format requirements summarized on the following checklist: (<https://www.nature.com/documents/commsj-phys-style-formatting-checklist-article.pdf>) and formatting guide (<https://www.nature.com/documents/commsj-phys-style-formatting-guide-accept.pdf>).

REVIEWER COMMENTS:

Reviewer #1 (Remarks to the Author):

Review comments on the manuscript titled
“Detailed observations reveal the genesis and dynamics of destructive debris-flow surges”
by Aaron et al

This study examines the mechanics of debris-flow surge formation and propagation using a combination of high-resolution field measurements and depth-averaged numerical modeling. The research aims to characterize the initiation and evolution of debris flow surges. The authors mainly investigated the basal friction and the role of coarse-grained particles in modulating debris flow behavior. Field data, which capture the flow depth, velocity, and morphology, are systematically analyzed to identify key processes governing surge formation. Complementary numerical simulations are conducted to reproduce observed surge features. Findings from this research provide new insights into the formation and evolution of surges and contribute to advancing predictive modeling frameworks for debris-flow hazard assessment. The manuscript is logically structured. However, I believe there are still margins for improvement by considering my following comments. Therefore, I recommend a Major Revision before the manuscript be considered for publication.

In the section “Discharge and Basal Friction”

Lines 129–131: The authors suggest that the presence of boulders leads to higher-frequency fluctuations in basal resistance at CD27 compared to CD29. Could the available video recordings provide supporting evidence for this interpretation? Additionally, considering that CD29 is located over 1 km downstream of CD27, is there any indication that these boulders are deposited before reaching CD29?

Figure (3): The inverted basal resistance exhibits high-frequency fluctuations at all monitored cross sections. However, in the subsequent numerical simulations, the non-monotonic friction model introduced in the Methods section was applied, which does not account for these fluctuations. Therefore, what impact might these fluctuations have on the numerical results? Specifically, do these high-frequency fluctuations in the basal friction coefficient affect the initiation and evolution of surges within the flow?

In the section “Surge Formation and Propagation”

Line 150 – 153: I recommend that the authors provide a more detailed discussion on the concepts of “roll waves” and “erosion–deposition waves,” especially regarding how the dominance of these two wave types varies with key flow

properties. It would be valuable to clarify their relationships with flow depth, Froude number, and basal friction coefficient (Figure 4). Additionally, the authors are encouraged to explain how these insights can improve the understanding, prediction, and delineation of debris-flow hazards in natural channels. Given that a change of slope as small as 0.8 degree would result in the transition of surge regimes. I am curious whether the surge regimes will be highly uncertain and might not help the delineation of debris flow process across complex terrains.

In the section "Implication for Debris flow Hazard and Mechanics"

I strongly recommend that the authors expand their discussion regarding whether the single-phase model can adequately capture the complex kinematic characteristics of debris flows. Specifically, the authors attribute the observed fluctuations in the basal friction coefficient to the presence of boulders; however, the subsequent numerical investigations employ a single-phase model, which inherently neglects the influence of large boulders and the intricate solid-fluid interactions. How do the authors justify that the effects of boulders on surge formation and propagation are negligible within this modeling framework? A more detailed explanation addressing this inconsistency would be valuable.

Moreover, the pioneer work by Hsu et al. (2014) and Song & Choi (2021) demonstrate that debris flow-induced bed erosion is predominantly governed by collisional stresses transmitted from solid particles onto the bed material. These findings imply that particle-induced erosion can significantly influence the erosion-deposition dynamics of debris flows and potentially influence the erosion-deposition waves. I strongly encourage the authors to discuss the implications of neglecting such multi-phase interactions and their potential impact on the modeled surge behavior.

This study demonstrates that the single-phase model can successfully reproduce debris flow kinematics through inverse analysis by calibrating an effective basal friction coefficient. While this is an important achievement, I encourage the authors to further discuss the model's predictive capability. In particular, without accounting for the multi-phase nature of debris flows, how can one determine the appropriate equivalent friction coefficient in advance for forward predictions? Given that the calibrated friction in the inverse analysis may implicitly incorporate multi-phase effects (such as particle-fluid interactions and collisional stresses), clarifying the physical meaning and applicability of this effective friction in predictive modeling would greatly enhance the practical relevance of the study.

Reference:

Hsu, L., Dietrich, W. E., & Sklar, L. S. (2014). Mean and fluctuating basal forces generated by granular flows: Laboratory observations in a large vertically rotating drum. *Journal of Geophysical Research: Earth Surface*, 119(6), 1283-1309.

Song, P., & Choi, C. E. (2021). Revealing the importance of capillary and collisional stresses on soil bed erosion induced by debris flows. *Journal of Geophysical Research: Earth Surface*, 126(5), e2020JF005930.

Reviewer #2 (Remarks to the Author):

This study demonstrates remarkably clear debris flow surges using valuable data obtained through advanced methods. It is an important study, including numerical simulations that successfully reproduce the surges. Initially, I suspected data processing issues due to the excessively sharp variation patterns in Fig. S3 and S6. However, these variations are supported by the supplementary videos (S5, though less clear in S4). Since such phenomena are difficult to replicate in flume tests due to the limitations of channel length, such data had not been previously obtained for debris flows. As outlined below, additional information is necessary, but I believe the study is worth publishing.

Reproducibility of Numerical Simulations

The numerical simulations appear to have good reproducibility (Fig. 2). However, from a practical standpoint, discharge is sometimes more fundamentally important than the roll waves. I suggest including the discharge simulation results in Fig. 3 for comparison with the observations.

Additionally, it would be beneficial to illustrate where and how the surges originated and developed between Gazoduc and CD27 in the calculations (if possible). Complementing the observational data with numerical simulations would be an effective way to verify the phenomenon.

The slope of Gazoduc should also be presented in L144. Even if obtaining an accurate value in the field is challenging, as noted in the Fig. 3 caption, the value used in the simulations should be provided. Furthermore, since channel slope is critical for debris flow behavior, as mentioned by the authors (L199), the paper should provide a more detailed explanation of the assumption that multiple weirs in the channel were ignored in setting the topographic conditions.

Cause of Surges

The authors attribute the formation of the surge waves to boulders, stating that the waves are generated by these boulders and evolve as they move downstream (e.g., L194-196). Upon reviewing the supplementary videos, I observed that at Gazoduc, CD27, and CD29, boulders accumulate in the front part (S1-S3), which is typical of debris flows. However, in the wave videos at CD27 and CD29 (S4 and S5), the particle size appears to be significantly finer. Given this, where are the boulders responsible for triggering these waves likely to be located?

Additionally, the relationship between boulders and flow friction is discussed in L125-132 and Fig. 3. However, since the surges are not prominent in the tail of the debris flow at CD27 but become more evident in the tail at CD29, this explanation may not be sufficient.

L168-170: I understand the authors' intended argument. However, the results in Fig. 3 can also be interpreted as indicating that a single hazardous peak discharge is divided into multiple, safer peaks due to the surge formation mechanism. More information should be added regarding the risk assessment of debris flows.

Time Discrepancy in Debris Flow Data:

The timestamps displayed in the supplementary video differ from those in Figs 2 and 3. Does this indicate that these

correspond to different debris flow events, or is there an error in one of the timestamps?

Reviewer #3 (Remarks to the Author):

General comments:

This manuscript ties high-resolution field observations of debris flows to simulated model results. Overall the manuscript is clearly written, and the figures are of high quality. It is easy to follow the main points of the manuscript and I see the value in this work. I have a few major suggestions that I think the authors may want to consider to make this into a paper with broader reach.

First, in the manuscript, you say that you use "measurements ...to calibrate...a numerical model". In the world of numerical modeling that can mean many things. It can mean that you use measurements to constrain parameter values. In this case, I think you did that with basal friction (line 142), but I can't tell how you decided on the other parameter values from the current text. Calibration can also simply mean that you have an objective function, and you tweak physically reasonable parameters until you reach some minimum error. In this case, I don't think you did that either. It seems like you had field measurements of depth and velocity and you had a model that output depth and velocity results that look visually similar. It's not clear to me that you iterated through model runs to minimize model versus observed errors. I do see that you clearly compare observed and simulated depth/Froude Number in Figure 4, but without any type of error analysis.

At the end of the manuscript in it's current form it is not clear to me what readers should take away from the field-model connection. I think the current take-away is supposed to be that you have a model that generates surges using estimates of basal friction constrained by field-estimates of discharge. But as you point out, other models generate surges, here are two that I'm aware of, and I'm sure there are many others.

McGuire, Luke A., Francis K. Rengers, Jason W. Kean, and Dennis M. Staley. "Debris flow initiation by runoff in a recently burned basin: Is grain-by-grain sediment bulking or en masse failure to blame?." *Geophysical Research Letters* 44, no. 14 (2017): 7310-7319.

Kean, J. W., McCoy, S. W., Tucker, G. E., Staley, D. M., & Coe, J. A. (2013). Runoff-generated debris flows: Observations and modeling of surge initiation, magnitude, and frequency. *Journal of Geophysical Research: Earth Surface*, 118(4), 2190-2207.

So my point is not to be overly critical, I think it's interesting and valuable work. Rather, I would challenge the authors to work a little harder to show how the field data are being used to calibrate the model. On line 147 you say that "...[the model is not] explicitly calibrated with these measurements". So maybe part of the issue I'm having is that there is disagreement between the initial statement about calibration in the abstract and subsequent text in the manuscript.

The second issue I would suggest the authors address head-on is the parameter calibration. There are 8 parameters in this model and it is not clear to me how these parameters were chosen. As these authors know well, using this model or any other model for future prediction is entirely dependent on appropriate parameter choice. If I was a future user of the model, this paper does not give me any clear guidance on how or why I would choose specific parameter values. I think any type of guidance on this would be incredibly valuable in the next version of the manuscript.

My remaining comments are in the line comments below.

Line comments:

13-14. I have a personal preference here that has been reinforced by my organization, that I don't like these types of grandiose superlative statements because they are rarely true.

23. Just a logic check here. This indicates that it was a series of surges and not just the first surge or one surge out of many that was the most destructive. Do the observations support the fact that the destruction was worse specifically because of multiple surges, or is it more accurate to say something like: "The debris flow was destructive and was observed to have multiple surges which may have each contributed to increased levels of destruction...". If you aren't able to truly attribute the source of destruction to surges specifically I don't think you should make it sound like that in the text.

26. Here is the danger in superlative statements. For example, Rapstine 2020 recorded 3D data at a speed of 29 Hz. But of course that was in an outdoor flume and focused on initiation. Rengers 2021 recorded 1D data at 60 Hz, again at an outdoor flume. I'm not saying these are exactly comparable, but when you start using language like 'highest resolution...ever made' the reality is that to make that claim you have to add qualifying language to make sure that the statement is true. My suggestion is just to avoid the grand statements so you don't have to spend time qualifying it.

Rapstine, T. D., Rengers, F. K., Allstadt, K. E., Iverson, R. M., Smith, J. B., Obryk, M. K., et al. (2020). Reconstructing the velocity and deformation of a rapid landslide using multiview video. *Journal of Geophysical Research: Earth Surface*, 125, e2019JF005348. <https://doi.org/10.1029/2019JF005348>

Rengers, F. K., Rapstine, T. D., Olsen, M., Allstadt, K. E., Iverson, R. M., Leshchinsky, B., ... & Smith, J. B. (2021). Using high sample rate lidar to measure debris-flow velocity and surface geometry. *Environmental & Engineering Geoscience*, 27(1), 113-126.

35. I don't think you mean that the wave is propagating through stationary debris. This makes me visualize something like creating a wave by picking up a rug and creating a wave that moves from one end of it to another. But here I think you mean something like waves propagating over or across initially stationary debris, right?

64. I haven't looked at the methods section yet, but it's not immediately clear to me how you get the depth-averaged velocity for any portion of the flow other than the first surge. And even if you restrict it to a single surge, if you have a mobile bed I don't know how you would truly get the depth-averaged velocity from lidar if you can't see the base of the mobile bed.

65. It's a little hard for me to visualize what you are saying here and in the caption from figure 2 where you are saying that the material velocity is slower than the wave velocity. Could you create a conceptual cartoon that demonstrates that with a few arrows to help readers. If you don't have room for more figures maybe this can just be another panel in figure 2. In figure 2 and in the text, I understand that the surges are fairly constant in time and space between stations. So I don't think you are saying that the surges are accelerating. Are you trying to say that if you pick the tops an initial wave (w1) and a subsequent

wave (w2) the depth-averaged velocity of the material between w1 and w2 is slower than the speed of either w1 or w2?
83. How did you manually map those?
103. Is the 'erosion-deposition' wave here equivalent to the 'sediment capacitor' described by Kean et al. 2013 (ref. 10)
106. Can some of this change in velocity be related to dewatering of the flow?
127. Since Figure 2A is your map, I'm not sure this is what you mean to refer to in this case to support your statement about basal resistance fluctuations being correlated to large boulders. Do you mean Figure 2B?
174. The word in this sentence that worries me a bit is 'significant'. I believe here you are talking about the changes between CD27 and CD28. In Figure 2C I see a depth that peaks around 2 m and decreases to about 0.75 m, and in 2D it peaks around 2.5 and decreases to about 1. Those seem different. The velocity in 2C peaks around 5 m/s and decreases to 0, but it looks pretty similar to 2D. It's unclear to me that the velocity changes are statistically significant, and even though the depth in 2C certainly looks to be like a different population of values than the values in 2D, I'm not sure how much a difference in ~0.5 m matters. I'm not saying it doesn't matter, I'm just saying it's not clear to me how important that is from the text.
179. I'm glad you acknowledged the boulders here because that seems to be like the most important piece of the observations, and so it is a little unsatisfying that it can't be explicitly incorporated into the model.
188. This is only true when it is properly parameterized. Guidance on how to correctly 'guess' those parameters would be helpful.
312. Can you say just a bit more about what you did to manually track these? For example, did you pick a static point and track how many frames it took for some observed object to move past the point?
328. I think the most important thing you can note here is if you have a bedrock bed, a shallow mobile sediment bed above bedrock, or a deep mobile sediment bed. If it's the 3rd choice, then I think that calls into question the ability to actually estimate a depth unless you have some good reasons to believe that a pre-flow channel filled with sediment wouldn't be mobile.
Equation 7. Consider using z to represent elevation rather than y.
389-390. I think it'd be helpful to provide guidance to future users on how they could come up with reasonable values here. Also, it seems like inherent in this sentence you must have done some sort of optimization scheme to determine those values. Any information on how users could determine values for these 8 free parameters would be helpful.
396. Please specify if you can always use the bed-slope here and if you would ever need to adjust this to the friction slope of a flow.
402. Maybe change 'illustrative values' to 'physically-reasonable values based on X' where X is some reason that makes sense. Either a prior study, or some reasonable constraint.
419. Is there any way to acknowledge the field observations on Figure 5?
References are incomplete. For example, just looking at 1 and 2 you don't list all the authors, you just use et. al.
Figure 1 caption. I think there are two important words that are wrong in this sentence: "Surge wave propagating through stationary material at CD 29." I don't think you mean through, I think you mean something like 'over'. Also, I think you are trying to say that the bed material is initially stationary prior to the surge that flows over it, but you are not claiming that the bed material remains stationary after the surge moves over it. So maybe just change to say 'initially stationary' or something like that to avoid confusion. My main point here is that you are showing a surge that moves across or over material that is initially stationary, but the text here doesn't exactly convey that, which could lead to confusion.
Also in the Figure 1 caption, I see a lot of inconsistency in how you write the cite names. I see CD27 on the map, (CD) 27 in the caption, (CD 27) in figure C, and CD 27 in the text. Suggest being more consistent with spaces and parens throughout.
Figures S1, S3, and S5, it seems a little strange that the orientation of flow changes between these figures. I'm sure there is a good reason, but for consistency would it be possible to make them all flow in the same direction, either toward the top or bottom of the page?

Communications Earth & Environment is committed to improving transparency in authorship. As part of our efforts in this direction, we are now requesting that all authors identified as 'corresponding author' create and link their Open Researcher and Contributor Identifier (ORCID) with their account on the Manuscript Tracking System prior to acceptance. ORCID helps the scientific community achieve unambiguous attribution of all scholarly contributions. You can create and link your ORCID from the home page of the Manuscript Tracking System by clicking on 'Modify my Springer Nature account' and following the instructions in the link below. Please also inform all co-authors that they can add their ORCIDs to their accounts and that they must do so prior to acceptance.
<https://www.springernature.com/gp/researchers/orcid/orcid-for-nature-research>

If you experience problems in linking your ORCID, please contact the Platform Support Helpdesk.

Version 1:

Decision Letter:

Dear Professor Aaron,

Your revised manuscript titled "Detailed observations reveal the genesis and dynamics of destructive debris-flow surges" has now been seen by our reviewers, whose comments appear below. In light of their advice we are delighted to say that we are happy, in principle, to publish a suitably revised version in Communications Earth & Environment.

We therefore invite you to revise your paper one last time to address the remaining concerns of our reviewers. At the same time we ask that you edit your manuscript to comply with our format requirements and to maximise the accessibility and therefore the impact of your work.

EDITORIAL REQUESTS:

*****Please take care to match our formatting and policy requirements. We will check revised manuscript and return manuscripts that do not comply. Such requests will lead to delays. *****

SUBMISSION INFORMATION:

OPEN ACCESS:

Communications Earth & Environment is a fully open access journal. Articles are made freely accessible on publication. For further information about article processing charges, open access funding, and advice and support from Nature Research, please visit <https://www.nature.com/commsenv/open-access>

Link Redacted

Best regards,

Alireza Bahadori, PhD
Associate Editor
Communications Earth & Environment
Consulting Editor
Communications Sustainability

REVIEWERS' COMMENTS:

Reviewer #1 (Remarks to the Author):

Thanks for taking the time to address the comments that we provided. I think that the discussion is now enriched and the manuscript is a very useful contribution to the literature. I have no further comments.

Reviewer #2 (Remarks to the Author):

After reading the authors' reply and the revised manuscript, I found that all of my comments were sincerely addressed and that the manuscript has been improved to a publishable level.

I would like to thank the authors for adding Fig. 5, which clearly helps readers understand the nature of the debris flows focused on in this study. I like it very much. However, please confirm the time labels in the figure. The time intervals are not 90 seconds (as described in the added text at L195), and even assuming 90-second intervals, they do not appear to correspond with the time labels in Fig. 2.

I also thank the authors for showing me the comparison between observed and simulated flow discharges. This is exactly the kind of figure I had hoped to see. The comparison clearly shows that, while the numerical simulation successfully reproduces the travel time and flow surface patterns (Fig. 2), it has some difficulty reproducing the flow discharge. For several minutes after the debris-flow front, the simulated discharge appears to be less than half of the actual discharge, indicating that the simulated debris flow is stretched longitudinally. Additionally, at CD29, it is interesting to note that the friction coefficient obtained by the numerical simulation is considerably smaller than that estimated from observations (shown in the new Fig. 3). Although I am not requesting an additional figure, I feel that these pieces of information would be effective for readers to understand the study more deeply.

Thank you for this interesting paper.

Reviewer #3 (Remarks to the Author):

I think that the authors have done a nice job addressing my original comments. I don't have any additional major suggestions at this point.

I saw in their rebuttal that they were going to try to add a video to the supplemental, and I don't see any videos. But I imagine that they did upload that video and that there must just be something preventing me from seeing it. Similarly, I can't see the simulation you mention in this sentence: "The full simulation continues for approximately 15 minutes after the final snapshot and may be viewed as an animation provided in the Supplementary Material."

263: I'd change "debris-flow hazard" to "debris-flow hazards"

Figures 5, 6, and 7: Can you save these figures at a higher resolution? They are blurry on my pdf.

REVIEWER COMMENTS:

Reviewer #1 (Remarks to the Author):

Review comments on the manuscript titled

“Detailed observations reveal the genesis and dynamics of destructive debris-flow surges”

by Aaron et al

This study examines the mechanics of debris-flow surge formation and propagation using a combination of high-resolution field measurements and depth-averaged numerical modeling. The research aims to characterize the initiation and evolution of debris flow surges. The authors mainly investigated the basal friction and the role of coarse-grained particles in modulating debris flow behavior. Field data, which capture the flow depth, velocity, and morphology, are systematically analyzed to identify key processes governing surge formation. Complementary numerical simulations are conducted to reproduce observed surge features. Findings from this research provide new insights into the formation and evolution of surges and contribute to advancing predictive modeling frameworks for debris-flow hazard assessment. The manuscript is logically structured. However, I believe there are still margins for improvement by considering my following comments. Therefore, I recommend a Major Revision before the manuscript be considered for publication.

Thanks very much for your thorough review and detailed comments. We really appreciate you taking the time and think that your feedback has thoroughly improved the manuscript.

In the section “Discharge and Basal Friction”

Lines 129–131: The authors suggest that the presence of boulders leads to higher-frequency fluctuations in basal resistance at CD27 compared to CD29. Could the available video recordings provide supporting evidence for this interpretation? Additionally, considering that CD29 is located over 1 km downstream of CD27, is there any indication that these boulders are deposited before reaching CD29?

This is a really interesting comment. We have systematically mapped boulder occurrence in this event at the three stations, using manual labelling, object-detection based machine vision, and video interpretation. There are many fewer boulders at CD29. This may be due to increased flow depth at this station, however we do think it is indicative, and demonstrates the point you are making. Regarding deposition, we have examined UAV ortho images from before and after the event. We do see some boulder deposition on these, but it is difficult to systematically know the origin of the deposited features.

We have updated line 144 to say:

We observe fewer boulders at CD 29 than CD 27, which may be due to boulder deposition between the two stations, or because boulders become obscure when the flow depth rises.

Figure (3): The inverted basal resistance exhibits high-frequency fluctuations at all monitored cross sections. However, in the subsequent numerical simulations, the non-monotonic friction model introduced in the Methods section was applied, which does not account for these fluctuations. Therefore, what impact might these fluctuations have on the numerical results? Specifically, do

these high-frequency fluctuations in the basal friction coefficient affect the initiation and evolution of surges within the flow?

This is a really insightful point, which we think points to a broader revision we have made in the manuscript, which is to better highlight the goals of the modelling, and better address uncertainties. With the modelling, we aim to include the minimal physics needed to usefully reproduce our field observations. In this context, our approach is roughly equivalent to ‘averaging over’ the smaller fluctuations to arrive at a bulk effective frictional response that nevertheless produces good agreement with the subsequent surges, which we think provides strong support for our analysis of surge generation and propagation.

The high frequency fluctuations you note are the result of boulders within the flow, as well as methodological uncertainties with estimating spatial and temporal derivatives from our data. In response to your final question, it seems plausible to us that the upstream fluctuations due to boulders could promote wave development by generating disturbances that develop downstream into roll waves and we have alluded to this in the revised manuscript. However, more data (and further analysis) would be required to isolate this effect and reach a firmer conclusion.

We have updated the line 463 of the methods to say:

It should be noted that estimating derivatives from the field data results in some high-frequency noise, which is apparent in the results of the inversion procedure.

We have updated the main text with the following paragraphs:

*The numerical modelling results demonstrate that the formation and coarsening of surge waves can be quantitatively captured using a single-phase 1D depth-averaged model, so long as the friction law and channel slope angles are appropriately selected (Figure 2 and Figure 4). Given the shallow gradients of the Illgraben fan, a change in mean slope angle of just 1° leads to a measurable change in flow depth and velocity (Figure 2 and Figure 4) that is commensurate with the field data, thereby implying that debris-flow dynamics can be sensitive to seemingly gradual topographic variations⁴⁵. Our simulations cannot capture all the details of the flow, in particular the front shape and behaviour at the tail. This is likely due to the fact that various time-varying properties of the liquefied slurry^{18,46–48} are not accounted for. **For example, the field measurements at Gazoduc and CD 27 demonstrate that large boulders locally increase the basal resistance. This leads to the presence of relatively greater frictional variations over the length of the bouldery front, and unsteady flow behind the front, consistent with observations made by other researchers^{9,16,17,49}.***

*In summary, our results provide a new perspective on debris-flow motion. The field measurements show that field-scale debris flows are spatio-temporally complex, and can transition between different flow and surge regimes as they move down a fan. These complex dynamics directly control debris-flow destructiveness, and therefore must be accounted for when managing debris-flow hazard. **Remarkably, we find that the measured phenomenological complexity is governed by two main mechanisms: increased flow resistance due to the presence of large boulders, which cause highly unsteady velocities, and the development of surface instabilities into damaging waves. While there is likely some coupling between these two mechanisms, with large boulders generating surface disturbances that grow into surge waves, we conclude that the linear instability can occur without explicitly accounting for the effects of large boulders.** Furthermore, we show that the instability can be captured using a single-phase numerical model which quantitatively reproduces spatio-temporal variations in surge velocities, depths and types, and reveals that debris-flow dynamics can be acutely sensitive to the channel slope angle. The striking*

effectiveness of our relatively simple modelling approach suggests that suitably parameterized, computationally inexpensive models could provide a promising path towards tools for markedly improved debris-flow hazard assessment. Incorporating the effects of composition variation and in particular, the influence of large particles^{56,57}, could lead to yet greater fidelity.

In the section “Surge Formation and Propagation”

Line 150 – 153: I recommend that the authors provide a more detailed discussion on the concepts of “roll waves” and “erosion–deposition waves,” especially regarding how the dominance of these two wave types varies with key flow properties. It would be valuable to clarify their relationships with flow depth, Froude number, and basal friction coefficient (Figure 4). Additionally, the authors are encouraged to explain how these insights can improve the understanding, prediction, and delineation of debris-flow hazards in natural channels. Given that a change of slope as small as 0.8 degree would results in the transition of surge regimes. I am curious whether the surge regimes will be highly uncertain and might not help the delineation of debris flow process across complex terrains.

This is a fantastic comment. We have updated the introduction to include more details concerning the existing understanding of these waves:

The dynamics of debris flows are strongly coupled to the frictional resistance that they experience at their base. No consensus has yet emerged about what controls debris-flow friction. However, it has been suggested that it is affected by coarse grained components within the flow^{9,16,17}, pore pressure effects^{9,18–20}, as well as flow depth and velocity^{4,9,11,12,21–24}. Debris-flow surges can manifest as regular trains of quasi-steady travelling waves (‘surge waves’) that have been hypothesised¹³ to occur for the same essential reason as the phenomenon of ‘roll waves’ found in shallow flows of laminar²⁵ and turbulent water^{26,27}, as well as dry granular media^{21,22} and many other complex fluids^{28–30}. However, obtaining compelling evidence in support of this view requires measurements of the spatiotemporal development of surges, together with a characterisation of the debris flow friction.

Roll waves are characterised by downstream propagating undulations of the flow surface led by a steep shock in which discharge is preferentially concentrated. They grow spontaneously from small perturbations due to an instability that is driven by gravity and the resultant frictional feedback of the flowing material¹. A closely related class of pulses, termed ‘erosion-deposition waves’ can occur in materials featuring an effective yield stress, such as dry granular flows¹². These propagate through recently deposited debris, by mobilising a layer of recently-deposited debris en-masse at their fronts and redepositing it at their trailing edge. This mechanism is distinct from the typical view of debris-flow erosion, which relates to mobilization of the static bed material along the path due to collisional stresses transmitted from the flowing debris to the bed material^{31,32}. For flows whose frictional properties are well constrained, detailed predictions of the size, shape and speed of both wave types can be made^{2,5,6}. By contrast, the difficulty of obtaining direct quantitative field measurements of full-scale debris flows has left the rheology of these flows underdetermined, thereby limiting our ability to understand their surge development, and reliably model the associated hazards.

However, after reflecting on your comments, we feel that the delineation of these classes is not always helpful, since they do not carry substantially different hazard implications and can coexist in broadly the same regions of parameter space. The erosion-deposition waves observed in the flow are the result of roll waves fully maturing by reaching their maximum amplitude. While the

transition between the two classes is aided by the deceleration that occurs as waves approach the shallower slopes of CD 29, we would also expect to see erosion-deposition waves (for example) at CD 27, if wave development had been triggered earlier (thereby enabling waves to fully mature further upstream in the channel). Indeed, this has been observed in other flows at the Illgraben. Consequently, in the discussion section, we have de-emphasized the different surge wave classes somewhat, in order to better focus on the more fundamental finding from the work, which is a new perspective on debris-flow friction, and that the resulting macroscopic wave behaviour is the result of an instability present in this friction law:

A key implication of our findings is that debris flow surges can emerge spontaneously as the result of a linear instability, and that the resulting surge waves can manifest as either roll waves or erosion-deposition waves. This mechanism has been hypothesized by previous researchers^{11,13}, however direct high-resolution measurements with which to test these hypotheses have been lacking. Furthermore, no consensus on the correct rheological description for modelling debris flows exists, which is critical for effectively capturing roll wave instabilities in numerical models. The close agreement between our data and the numerical simulations both supports the existence of the underlying instability in the field, and provides empirical support to our tested friction law. Interestingly, analysis of this friction law, with the parameters in the presented simulation, implies that the flow is unstable regardless of Froude number, and that this condition is required for surge waves to form, regardless of compositional variations (see Methods). However, the (linear) spatial growth rate of the resulting surges is highly dependent on this quantity. In particular, the dependence is non-monotonic, reaching a maximum at intermediate values and ultimately decaying to zero as Froude numbers becomes very high⁵⁵. Our data and simulations support this interpretation, as shown in Figure 5, as well as Figure 2B, which demonstrates that no surge waves are present at Gazoduc until the flow depth and velocity have waned, and then small surges start to appear in the flow after about 10 minutes (see also Supplementary Figure 3). Similarly, Figure 5C and D show a period of increased instability as the source flux at Gazoduc wanes in the simulation. This suggests that the high discharge surges documented in the present event become most pronounced at intermediate Froude numbers, and would not have arisen in far faster, or far slower flows of the same composition. Continued application of theoretical insights of this kind, which can draw upon an extensive history of roll wave analyses in other systems, may soon help to disentangle open questions in debris flow science, such as why some events produce substantial surge trains, while others do not.

In the section “Implication for Debris flow Hazard and Mechanics”

I strongly recommend that the authors expand their discussion regarding whether the single-phase model can adequately capture the complex kinematic characteristics of debris flows. Specifically, the authors attribute the observed fluctuations in the basal friction coefficient to the presence of boulders; however, the subsequent numerical investigations employ a single-phase model, which inherently neglects the influence of large boulders and the intricate solid-fluid interactions. How do the authors justify that the effects of boulders on surge formation and propagation are negligible within this modeling framework? A more detailed explanation addressing this inconsistency would be valuable.

Moreover, the pioneer work by Hsu et al. (2014) and Song & Choi (2021) demonstrate that debris flow-induced bed erosion is predominantly governed by collisional stresses transmitted from solid particles onto the bed material. These findings imply that particle-induced erosion can significantly

influence the erosion-deposition dynamics of debris flows and potentially influence the erosion-deposition waves. I strongly encourage the authors to discuss the implications of neglecting such multi-phase interactions and their potential impact on the modeled surge behavior.

We completely agree that this was unclear in the previous version of the manuscript. To address this important comment, we first updated the text to better clarify the goals of the modelling, by adding the following text:

*To understand the mechanisms governing surge dynamics, we performed numerical simulations of the same flow equations used in the friction coefficient inversion procedure, with the 'Kestrel' open-source shallow flow software⁴¹ (Figures 2E-G). **Our approach here is to provide insights into the formation and dynamics of the measured surges while including the minimal amount of complexity, rather than to extensively calibrate a hazard model. We therefore used an existing granular friction law^{12,42,43} adjusted to quantitatively approximate the inferred field values of basal friction across the physical regimes recorded during the event (Figure 4), and use a simplified 1D topography that smoothly transitions between the measured slope angles at Gazoduc and CD 27 (4.5°) and CD 29 (3.5°).** Details of the model friction law and the process of parameter selection are given in the Methods section. As discussed later, the effects of large boulders on friction are not explicitly included in our simulations.*

*Waves spontaneously arise in the simulation from an underlying roll wave instability^{24,42} that is present for all flow conditions with the model friction law. **The simulations are effective at reproducing the observed depths and velocities at each measurement station, despite not being explicitly calibrated against these measurements (Figure 2), and trace out essentially the same regions of depth-Froude number space (black lines on Figure 4) as the corresponding values from the directly measured surges (coloured points on Figure 4). The data separation in Figure 4 demonstrates that the relatively modest change in slope between the two stations (1°) influences the transition of the observed surges between two regimes, of faster, thinner roll waves at CD 27 and slower, thicker erosion-deposition waves at CD 29 (Figure 2 and Figure 4).** For comparison, a simulation was performed using the same conditions, on a constant 4.5° slope. This undergoes the same instability development, ultimately leading to erosion-deposition waves, but lacks the clear separation between the CD 27 and CD 29 data observed in the field (see Supplementary Figure S 12).*

Next, we added the following text to the discussion to distinguish the role of boulders and surges in our flow:

*The numerical modelling results demonstrate that the formation and coarsening of surge waves can be quantitatively captured using a single-phase 1D depth-averaged model, so long as the friction law and channel slope angles are appropriately selected (Figure 2 and Figure 4). Given the shallow gradients of the Illgraben fan, a change in mean slope angle of just 1° leads to a measurable change in flow depth and velocity (Figure 2 and Figure 4) that is commensurate with the field data, thereby implying that debris-flow dynamics can be sensitive to seemingly gradual topographic variations⁴⁵. **Our simulations cannot capture all the details of the flow, in particular the front shape and behaviour at the tail. This is likely due to the fact that various time-varying properties of the liquefied slurry^{18,46-48} are not accounted for. For example, the field measurements at Gazoduc and CD 27 demonstrate that large boulders locally increase the basal resistance. This leads to the presence of relatively greater frictional variations over the length of the bouldery front, and unsteady flow behind the front, consistent with observations made by other researchers^{9,16,17,49}.***

*Additionally, pore pressure generation and dissipation is known to play a significant role in governing debris-flow friction^{18,48,50–53}. Our results show that low bulk basal friction angles, on the order of 3° to 6° degrees occur throughout the flow (Figure 3C and D), which imply significant pore pressures developing in the flowing debris. Nevertheless, we are able to simulate many bulk characteristics of the flow without explicitly accounting for spatial and temporal variations in pore pressure^{50,51,54} (Figure 2). **Consequently, we conclude that our newly developed friction inversion procedure is able to aggregate complex multi-phase debris flow physics into an apparent friction coefficient that can be implemented in single-phase depth-averaged models. While we have only applied the new methodology to a single event, our analysis provides the foundation for a more generalized, practical model that avoids the added complications and computational cost inherent in multi-phase and non-depth-averaged systems. This should become possible once the analysis procedure has been applied to a large number of debris-flow events.***

Regarding multi-phase interactions influencing erosion and deposition in debris flows, we again agree with the reviewer that this is critical in debris flow motion. However, the success of our relatively simple model indicates that these interactions may not govern this phenomenon in all cases, particularly when surge waves are present. Furthermore, at our study site there are checkdams which stabilize the bed and suppress erosion. In fact, if we integrate the discharge timeseries at the various stations we get remarkably similar volumes (~21,000 m³ at Gazoduc, and ~22,000 m³ at CD 27, with the differences likely in the range of uncertainties surrounding this estimate). We highlight the following text which was presented above as an addition to the introduction:

*Roll waves are characterised by downstream propagating undulations of the flow surface led by a steep shock in which discharge is preferentially concentrated. They grow spontaneously from small perturbations due to an instability that is driven by gravity and the resultant frictional feedback of the flowing material¹. A closely related class of pulses, termed ‘erosion-deposition waves’ can occur in materials featuring an effective yield stress, such as dry granular flow². These propagate through recently deposited debris, by mobilising material at their fronts and re-depositing it at their trailing edges. **It should be noted that this is a distinct mechanism from the typical view of debris-flow erosion, which relates to mobilization of the static bed material along the path due to collisional stresses transmitted from the flowing debris to the bed material^{3,4}.** For flows whose frictional properties are well constrained, detailed predictions of the size, shape and speed of both wave types can be made^{2,5,6}. By contrast, the difficulty of obtaining direct quantitative field measurements of full-scale debris flows has left the rheology of these flows underdetermined, thereby limiting our ability to understand their surge development, and reliably model the associated hazards.*

This study demonstrates that the single-phase model can successfully reproduce debris flow kinematics through inverse analysis by calibrating an effective basal friction coefficient. While this is an important achievement, I encourage the authors to further discuss the model’s predictive capability. In particular, without accounting for the multi-phase nature of debris flows, how can one determine the appropriate equivalent friction coefficient in advance for forward predictions? Given that the calibrated friction in the inverse analysis may implicitly incorporate multi-phase effects (such as particle-fluid interactions and collisional stresses), clarifying the physical meaning and applicability of this effective friction in predictive modeling would greatly enhance the practical relevance of the study.

Thanks for this nice comment, we agree that the practical implications of the study could be better highlighted. In general, we are a bit cautious to over generalize our findings, as they are based on one event and we would need a larger dataset before recommending a true prediction framework,

which is the subject of ongoing work. However, we remain confident about the practical implications of our approach. Though it seems unlikely that a single effective friction rule will work well for all flows, it may be the case that different ‘classes’ of events with similar compositions could be characterised by separate single-phase bulk models, which would still represent a considerable reduction in complexity – surely valuable from a forward modelling perspective. We have updated the discussion to clarify this (see above), as well as added the following text to line 230:

While we have only applied the new methodology to a single event, our analysis provides the foundation for a more generalized, practical model that avoids the added complications and computational cost inherent in multi-phase and non-depth-averaged systems. This should become possible once the analysis procedure has been applied to a large number of debris-flow events.

We have further added the text given above to better clarify our numerical modelling goals, and expanded the supplementary information to better communicate how parameters were selected and demonstrate that other approximations to the friction function can also work well to justify the conclusions reached in the main text. This is what we mean when we say the model is not ‘extensively calibrated’ – a property we consider to be advantageous in this context, since it shows that the existence and development of surges is not overly contingent on the choices made:

The parameters used in our numerical model do not directly correspond to quantities that are measurable in the field, or obtainable from current experimental data. Therefore, Eq. [8] in the main text may be viewed as a phenomenological construction that is flexible enough to approximate the observed (reconstructed) friction values and extend them to a plausible model friction function valid for all $Fr, h > 0$. It is possible to devise many different sets of parameters to those in Table 1 that approximate the friction values constrained by observations. This allows for a wide range of parameter choices that can quantitatively reproduce the essential dynamics of the instability and wave growth discussed in the main text, as shown in Figure S 11.

Nevertheless, there are some useful theoretical considerations that can be used to constrain the selection of parameters, which we detail below. Since the purpose of the simulation in this paper is to validate the reconstructed friction values and establish the relevance of the underlying roll wave instability, we did not perform extensive parameter fitting beyond this. Instead, we conducted several simulations (of which the Figure S 11 cases form a subset) to verify the robustness of our conclusions to variations in the parameters and chose to present the result with the Table 1 values as a representative example.

Figure S 11: Robustness of numerical results to variations in the frictional parameters. Each panel shows contours of friction, simulated Fr versus h at CD 27 (red) and CD 29 (purple), and the steady balances of Eq. [1] for $\theta = 4.5^\circ$ (lower dashed lines) and $\theta = 3.5^\circ$ (upper dashed lines). Panel A is the simulation presented in the main text, using the parameters from Table 1, while panels B and C

use the following modified values: (B) $\mu_1 = -0.025, \mu_2 = 0.12, \beta = 0.5, \beta_* = 0.15, \kappa = 0.2$; (C) $\mu_1 = 0.02, \mu_2 = 0.5, \beta = 8, \beta_* = 0.1, \kappa = 0.01, \Gamma = 1$. Note that the three cases lead to friction functions that are relatively close to the observed friction where data is available, but diverge from one another outside this region. The colorbar shows the friction coefficient.

Three parameters in the friction law, μ_3, L, κ control the behaviour of the friction law near the onset of deposition ($Fr < \beta_*$) and do not play a role in the roll wave instability. The value of μ_3 and the length scale L dictate the range of uniform flows that may exist as static deposits at a given slope angle. Specifically, an initially uniform static deposit of depth h can only remain stationary if $\mu_{\text{start}}(h) > \tan \theta$, which rearranges to give

$$h < L \left(\frac{\mu_2 - \mu_1}{\tan \theta - \mu_3} - 1 \right).$$

Since we know that static deposits did form at CD 29, this places a constraint on μ_3 and L (given μ_1 and μ_2 , which are dependent on considerations below). However, without additional data, such as the yield strength of the debris, this does not fully determine these parameters. Therefore, we chose some physically plausible values and verified that simulations are qualitatively insensitive to reasonable adjustments of these selections. (For reference, using our chosen values in the inequality above implies that uniform static layers can form for $h < 1.24\text{m}$ at CD 27 and $h < 2.16\text{m}$ at CD 29.)

The variable κ controls the behaviour of the low Fr frictional regime, which governs the onset of flow arrest and is only accessed intermittently by the waves that form deposits at their tails. A higher value of κ causes deposits to form more rapidly whenever they are triggered and therefore influences the shape of erosion-deposition waves. We set it to a value that produced waves with reasonable qualitative resemblance to the observed profiles at CD 29. Roll waves transition into erosion-deposition waves when their tails drop below $Fr < \beta_*$. Higher values of β_* somewhat enhance the phenomenological distinction between the two wave classes by allowing erosion-deposition waves to form more readily. Our chosen value ($\beta_* = 0.2$) is large enough to promote the formation of clear erosion-deposition waves (as observed in the field), but not so large that it interferes with the roll wave instability that occurs at higher Fr .

The remaining four parameters, determine the friction values in the dynamic regime ($Fr > \beta_*$), within which the initiation and development of roll waves are observed in the field data. Roll waves emerge from a linear instability of steady uniform flow. Under steady uniform conditions, gravitational forcing is exactly balanced by frictional resistance:

$$\mu_b(Fr, h) = \tan \theta \quad [S1]$$

which, for each slope angle θ , defines a curve in (Fr, h) space. When $Fr > \beta_*$ this may be rearranged to give the linear relationship

$$h = \left(\frac{L}{\beta} \cdot \frac{\mu_2 - \tan \theta}{\tan \theta - \mu_1} \right) (Fr + \Gamma), \quad [S2]$$

Since, h and Fr are both positive, $L/\beta > 0$, and $\tan^{-1} \mu_1$ and $\tan^{-1} \mu_2$ are respectively interpreted as minimum and maximum slope angles upon which uniform steady flows can exist. Therefore, $0 < \tan^{-1} \mu_1 < \theta < \tan^{-1} \mu_2$ (although even permitting $\mu_1 < 0$ can produce a reasonable fit to the reconstructed friction, as demonstrated in Figure S 11B).

To restrict these parameters further, we note that in their idealised description as steady, stable travelling wave solutions to Eqs. [4] and [5] (main text), roll waves are mathematically constrained to contain flow with the values (Fr_0, h_0) at a 'critical point' somewhere along their length, where Fr_0 and h_0 are the Froude number and depth of the corresponding uniform flow that gave rise to them via linear instability³⁸. In other words, at a given slope angle, all roll waves must pass through the line defined in Eq. [S2] and the location of this intersection is contingent on the upstream flux (which

controls Fr_0, h_0). This can be seen in the simulation data of Figure S 11, which are 'pinned' to the black dashed lines. The field data in Figure 4 (main text) are clearly separated into quantitatively different regimes, consistent with the different mean slope angles at the two downstream measurement stations. Therefore, we selected μ_1, μ_2, β , and Γ so that the Eq. [4] lines pass through the field data at CD 27 ($\theta = 4.5^\circ$) and CD 29 ($\theta = 3.5^\circ$). Note that given a systematic procedure for deciding on optimal intersection points, wave data at three different slope angles would be required to uniquely determine μ_1, μ_2, β and Γ . Therefore, we choose illustrative values, which lead to good agreement with the inferred field values of friction, within the available data ranges, as demonstrated in Figure 4 (main text).

Reference:

Hsu, L., Dietrich, W. E., & Sklar, L. S. (2014). Mean and fluctuating basal forces generated by granular flows: Laboratory observations in a large vertically rotating drum. *Journal of Geophysical Research: Earth Surface*, 119(6), 1283-1309.

Song, P., & Choi, C. E. (2021). Revealing the importance of capillary and collisional stresses on soil bed erosion induced by debris flows. *Journal of Geophysical Research: Earth Surface*, 126(5), e2020JF005930.

Reviewer #2 (Remarks to the Author):

This study demonstrates remarkably clear debris flow surges using valuable data obtained through advanced methods. It is an important study, including numerical simulations that successfully reproduce the surges.

We are very happy to read that you see the value of our study.

Initially, I suspected data processing issues due to the excessively sharp variation patterns in Fig. S3 and S6. However, these variations are supported by the supplementary videos (S5, though less clear in S4).

We shared the reviewers surprise when we first processed the data for this event. It motivated our study as we became convinced that these observations were of practical importance, and that we had a unique dataset with which to understand them.

Since such phenomena are difficult to replicate in flume tests due to the limitations of channel length, such data had not been previously obtained for debris flows.

As outlined below, additional information is necessary, but I believe the study is worth publishing.

Reproducibility of Numerical Simulations

The numerical simulations appear to have good reproducibility (Fig. 2). However, from a practical standpoint, discharge is sometimes more fundamentally important than the roll waves. I suggest including the discharge simulation results in Fig. 3 for comparison with the observations.

Thanks for this comment, we carefully considered this, and made the Figure below to compare:

We find that, since we are using 1D simulations, the discharge is the product of the flow depth and velocity, scaled with a constant geometric factor. As we already show flow depth and velocity on Fig 2., we find this information a bit redundant, and it makes Fig. 3 more complicated to interpret. We have therefore decided to omit this in the manuscript and leave the Figure as is. We hope the reviewer agrees with this.

Additionally, it would be beneficial to illustrate where and how the surges originated and developed between Gazoduc and CD27 in the calculations (if possible). Complementing the observational data with numerical simulations would be an effective way to verify the phenomenon.

We agree, and have added the following text and new figure to present this information. On reflection, this greatly clarifies the manner in which waves develop from small disturbances for the reader:

The simulation and field data mutually indicate that the large amplitude surges at CD 27 and CD 29 are caused by the growth and coalescence of small instabilities as the flow moves down the fan. This process is clarified in Figure 5, which shows a sequence of snapshots of the simulated flow depth along the section between Gazoduc and CD 29, at 90 second time intervals. The initial thick (1.5 m) inflow at Gazoduc is approximately uniform upstream of the front (Figure 5A). As the flow propagates towards CD 27, small disturbances become visible (Figure 5B), which grow into substantial waves, reaching amplitudes of half a metre or more by the time they reach CD 27 (Figure 5C). Alongside this, the flow thins due to the waning of the upstream flux, and briefly becomes significantly more unstable, as evidenced by the appearance of larger waves near Gazoduc (Figures 5C and 5D). Furthermore, the mature roll waves velocities are much faster than the material and front velocities (Figure 2), in line with the field observations. For example, the wave highlighted in solid red gains roughly 400 m relative to the front position in the 90 seconds between Figures 5C and 5D. Just prior to the snapshot in Figure 5E, it catches the front (which decelerates on the shallower gradients at CD 29) leaving a large deposit (blue shaded region on Figure 5E). The later panels (Figures 5D-F) contain four static deposits, indicated with blue shading. An erosion-deposition wave, separated by two such regions, can be seen travelling between CD 27 and CD 29 (Figure 5F). The full

simulation continues for approximately 15 minutes after the final snapshot and may be viewed as an animation provided in the Supplementary Material.

Figure 5: Snapshots of simulated flow depth versus the downslope distance from Gazoduc, separated by 90 second intervals. An individual waveform is highlighted with a solid red line in panels C & D to visualise its propagation and development relative to the rest of the flow. Grey dashed lines indicate the locations of the three measurement stations and regions of static deposit are shaded light blue.

The slope of Gazoduc should also be presented in L144. Even if obtaining an accurate value in the field is challenging, as noted in the Fig. 3 caption, the value used in the simulations should be provided. Furthermore, since channel slope is critical for debris flow behavior, as mentioned by the authors (L199), the paper should provide a more detailed explanation of the assumption that multiple weirs in the channel were ignored in setting the topographic conditions.

We would like to sincerely thank the reviewer for this comment. Investigating it has revealed a bug in our processing script, where our slope angles at CD27 and CD 29 were rotated by $\sim 1^\circ$. It was a systematic error, so the conclusions of the study remain valid, however we have changed the slope angle and redone all numerical modelling simulations. We have updated all relevant text and Figures.

Regarding ignoring the check dams, we have updated the text to better clarify the goals of the modelling, which were to keep complexity simple while still reproducing the bulk flow character. We simplified the topography because, on the scale of our model, the checkdams are relatively minor features:

Figure 3: Channel cross section spanning from Gazoduc to CD 29. Topographic data: from de Haas, T. *et al.* Flow and Bed Conditions Jointly Control Debris-Flow Erosion and Bulking. *Geophys. Res. Lett.* **49**, e2021GL097611 (2022).

We further investigated flow velocities immediately upstream and downstream of the checkdam at CD27, and found them broadly similar:

Figure 4: Velocity upstream (blue) and downstream (orange) of checkdam 27. Differences in flow velocity are minor.

We therefore think it is justified to neglect the influence of the checkdam in our simulations. Our interpretation is that the checkdam stabilizes the channel bottom, however once the flow has gone over the checkdam and flowed for a few tens of meters the effect on flow dynamics is largely lost.

Cause of Surges

The authors attribute the formation of the surge waves to boulders, stating that the waves are generated by these boulders and evolve as they move downstream (e.g., L194-196).

Through this and your other excellent comments we have realized that we had written a stronger causal relation between boulders and surge waves than we intended. In general we had hoped to present these as two distinct yet related mechanisms. The surge waves are formed from the growth and coalescence of instabilities on the free surface, governed by the friction law. In contrast, the boulders influence the basal friction, and generate surface fluctuations that grow by the linear instability mechanism, but are not required to generate these waves. We have clarified this in the introduction and discussion:

*In summary, our results provide a new perspective on debris-flow motion. The field measurements show that field-scale debris flows are spatio-temporally complex, and can transition between different flow and surge regimes as they move down a fan. These complex dynamics directly control debris-flow destructiveness, and therefore must be accounted for when managing debris-flow hazard. Remarkably, we find that the measured phenomenological complexity is governed by two main mechanisms: increased flow resistance due to the presence of large boulders, which cause highly unsteady velocities, and the development of surface instabilities into damaging waves. **While there is likely some coupling between these two mechanisms, with large boulders generating surface disturbances that grow into surge waves, we conclude that the linear instability can occur without explicitly accounting for the effects of large boulders.** Furthermore, we show that the instability can be captured using a single-phase numerical model which quantitatively reproduces spatio-temporal variations in surge velocities, depths and types, and reveals that debris-flow dynamics can be acutely sensitive to the channel slope angle. The striking effectiveness of our relatively simple modelling approach suggests that suitably parameterized, computationally inexpensive models could provide a promising path towards tools for markedly improved debris-flow hazard assessment. Incorporating the effects of composition variation and in particular, the influence of large particles, could lead to yet greater fidelity.*

However, in the wave videos at CD27 and CD29 (S4 and S5), the particle size appears to be significantly finer. Given this, where are the boulders responsible for triggering these waves likely to be located?

We agree, and hope that the clarification above has addressed this important point. In general, we think that either coarse particles are deposited between CD 27 and CD 29, or that they are obscured due to the larger flow depth at CD 29. We inspected post-event UAV imagery, but could not make definitive conclusions about this.

Additionally, the relationship between boulders and flow friction is discussed in L125-132 and Fig. 3. However, since the surges are not prominent in the tail of the debris flow at CD27 but become more evident in the tail at CD29, this explanation may not be sufficient.

We agree, and meant these as somewhat separate but coupled mechanisms. We hope that the text added above have clarified this point.

L168-170: I understand the authors' intended argument. However, the results in Fig. 3 can also be interpreted as indicating that a single hazardous peak discharge is divided into multiple, safer peaks due to the surge formation mechanism. More information should be added regarding the risk assessment of debris flows.

This is a really interesting comment, and we have thought about it in detail. We still think that our point stands, because the surge waves arise behind the front, and move with velocities that are faster than the front. They therefore serve to concentrate discharge, as opposed to divide it. This is consistent with prior analyses of roll waves in other media. We hope that the new Figure 5 better demonstrates this point, and we have been sure to point out that the surge waves travel faster than the front in the new text above.

Time Discrepancy in Debris Flow Data:

The timestamps displayed in the supplementary video differ from those in Figs 2 and 3. Does this indicate that these correspond to different debris flow events, or is there an error in one of the timestamps?

Thanks so much for your extremely thorough review! Yes, the timestamps of the LiDAR scanner are in local time, and the videos are UTC. We have noted this in the supplementary information.

Reviewer #3 (Remarks to the Author):

General comments:

This manuscript ties high-resolution field observations of debris flows to simulated model results. Overall the manuscript is clearly written, and the figures are of high quality. It is easy to follow the main points of the manuscript and I see the value in this work. I have a few major suggestions that I think the authors may want to consider to make this into a paper with broader reach.

First, in the manuscript, you say that you use “measurements ...to calibrate...a numerical model”. In the world of numerical modeling that can mean many things. It can mean that you use measurements to constrain parameter values. In this case, I think you did that with basal friction (line 142), but I can’t tell how you decided on the other parameter values from the current text. Calibration can also simply mean that you have an objective function, and you tweak physically reasonable parameters until you reach some minimum error. In this case, I don’t think you did that either. It seems like you had field measurements of depth and velocity and you had a model that output depth and velocity results that look visually similar. It’s not clear to me that you iterated through model runs to minimize model versus observed errors. I do see that you clearly compare observed and simulated depth/Froude Number in Figure 4, but without any type of error analysis.

Through this and other constructive comments we realized that we have not done a good job clarifying the goals of our numerical modelling analysis. First off, the particular sentence highlighted (“measurements ...to calibrate...a numerical model”) was insufficiently clear and has been removed.

As you correctly state below, there are many surge mechanisms proposed in literature. Our model captures one specific mechanism, which is the growth and coalescence of free surface instabilities. With our modelling, we wanted to test whether this mechanism can explain the observed surge development, as well as validate the reconstructed friction data. For this reason we avoided explicitly calibrating the model against velocity and depth, and instead only used the inferred basal slope angles, and the range of friction coefficients inverted from our data to construct an effective friction law that approximates these data. This process, which was dictated by theoretical considerations already outlined in the Methods section, has been substantially elaborated upon in the revised manuscript. As part of these revisions, we now make it clear that multiple simulations were conducted (see Supplementary Information), but for the purpose of confirming the robustness

of our primary conclusions, rather than to iterate towards an optimal set of parameters. It is our view that a more extensive calibration procedure that attempted to minimise error between observations and simulations would lead to overfitting the friction parameters to this single dataset.

Ultimately, we think the correspondence between our simulation and measurements strongly supports our hypothesized surge generation and propagation mechanism, and therefore makes other mechanisms less likely. We have updated the numerical modelling section with the following information:

*To understand the mechanisms governing surge dynamics, we performed numerical simulations of the same flow equations used in the friction coefficient inversion procedure, with the 'Kestrel' open-source shallow flow software⁴¹ (Figures 2E-G). **Our approach here is to provide insights into the formation and dynamics of the measured surges while including the minimal amount of complexity, rather than to extensively calibrate a hazard model. We therefore used an existing granular friction law^{12,42,43} adjusted to quantitatively approximate the inferred field values of basal friction across the physical regimes recorded during the event (Figure 4), and use a simplified 1D topography that smoothly transitions between the measured slope angles at Gazoduc and CD 27 (4.5°) and CD 29 (3.5°).** Details of the model friction law and the process of parameter selection are given in the Methods section. As discussed later, the effects of large boulders on friction are not explicitly included in our simulations.*

*Waves spontaneously arise in the simulation from an underlying roll wave instability^{24,42} that is present for all flow conditions with the model friction law. **The simulations are effective at reproducing the observed depths and velocities at each measurement station, despite not being explicitly calibrated against these measurements (Figure 2), and trace out essentially the same regions of depth-Froude number space (black lines on Figure 4) as the corresponding values from the directly measured surges (coloured points on Figure 4).** The data separation in Figure 4 demonstrates that the relatively modest change in slope between the two stations (1°) influences the transition of the observed surges between two regimes, of faster, thinner roll waves at CD 27 and slower, thicker erosion-deposition waves at CD 29 (Figure 2 and Figure 4). For comparison, a simulation was performed using the same conditions, on a constant 4.5° slope. This undergoes the same instability development, ultimately leading to erosion-deposition waves, but lacks the clear separation between the CD 27 and CD 29 data observed in the field (see Supplementary Figure S 12).*

At the end of the manuscript in its current form it is not clear to me what readers should take away from the field-model connection. I think the current take-away is supposed to be that you have a model that generates surges using estimates of basal friction constrained by field-estimates of discharge. But as you point out, other models generate surges, here are two that I'm aware of, and I'm sure there are many others.

Thanks for this comment, as above we have realised a need to better explain our modelling goals, and what they tell us. We think we have strong evidence that in the present event, and many others, a very specific mechanism is generating surges. This makes the other mechanisms much less likely to be the governing process, and allows us to understand these surge types in a broader context. In fact, the numerical modelling allows us to make the following broader implications:

A key implication of our findings is that debris flow surges can emerge spontaneously as the result of a linear instability, and that the resulting surge waves can manifest as either roll waves or erosion-deposition waves. This mechanism has been hypothesized by previous researchers^{11,13}, however direct high-resolution measurements with which to test these hypotheses have been lacking.

Furthermore, no consensus on the correct rheological description for modelling debris flows exists, which is critical for effectively capturing roll wave instabilities in numerical models. The close agreement between our data and the numerical simulations both supports the existence of the underlying instability in the field, and provides empirical support to our tested friction law. Interestingly, analysis of this friction law, with the parameters in the presented simulation, implies that the flow is unstable regardless of Froude number, and that this condition is required for surge waves to form, regardless of compositional variations (see Methods). However, the (linear) spatial growth rate of the resulting surges is highly dependent on this quantity. In particular, the dependence is non-monotonic, reaching a maximum at intermediate values and ultimately decaying to zero as Froude numbers becomes very high⁵⁵. Our data and simulations support this interpretation, as shown in Figure 5, as well as Figure 2B, which demonstrates that no surge waves are present at Gazoduc until the flow depth and velocity have waned, and then small surges start to appear in the flow after about 10 minutes (see also Supplementary Figure 3). Similarly, Figure 5C and D show a period of increased instability as the source flux at Gazoduc wanes in the simulation. This suggests that the high discharge surges documented in the present event become most pronounced at intermediate Froude numbers, and would not have arisen in far faster, or far slower flows of the same composition. Continued application of theoretical insights of this kind, which can draw upon an extensive history of roll wave analyses in other systems, may soon help to disentangle open questions in debris flow science, such as why some events produce substantial surge trains, while others do not.

McGuire, Luke A., Francis K. Rengers, Jason W. Kean, and Dennis M. Staley. "Debris flow initiation by runoff in a recently burned basin: Is grain-by-grain sediment bulking or en masse failure to blame?." *Geophysical Research Letters* 44, no. 14 (2017): 7310-7319.

Kean, J. W., McCoy, S. W., Tucker, G. E., Staley, D. M., & Coe, J. A. (2013). Runoff-generated debris flows: Observations and modeling of surge initiation, magnitude, and frequency. *Journal of Geophysical Research: Earth Surface*, 118(4), 2190-2207.

We would further like to note that one of the advantages of our approach is in the relative simplicity of using a single-phase description of the flow. Surges are generated in our model equations via a linear instability mechanism that is mathematically and physically well understood and common to a variety of different flowing media. If this mechanism is not present in our equations (see Figure 6) surges cannot persist. With this in mind, the purpose of presenting the simulation was first and foremost to tie the fundamental roll wave instability to observation data, which was not possible without high-resolution measurements recorded at multiple channel locations.

It is possible to add structural complexity to our model by including the motion of additional phases and/or bed evolution (morphodynamics), such as the examples in the cited references. It should be expected that such systems also support roll wave-like phenomena and potentially additional modes of instability. However, instabilities in models with this extra physics are far more challenging to study. When this has been done, it has typically revealed fundamental shortcomings in large classes of these models that render them without well-defined solutions. See for example, the following two references:

Chavarrías, V., Schielen, R., Ottevanger, W., & Blom, A. (2019). Ill posedness in modelling two-dimensional morphodynamic problems: effects of bed slope and secondary flow. *Journal of Fluid Mechanics*, 868, 461-500.

Langham, J., Woodhouse, M. J., Hogg, A. J., & Phillips, J. C. (2021). Linear stability of shallow morphodynamic flows. *Journal of Fluid Mechanics*, 916, A31.

Furthermore, we note that despite the added complexity, the models cited do not appear to do a substantially better job at quantitatively predicting observed surges than our simpler approach.

So my point is not to be overly critical, I think it's interesting and valuable work. Rather, I would challenge the authors to work a little harder to show how the field data are being used to calibrate the model. On line 147 you say that "...[the model is not] explicitly calibrated with these measurements". So maybe part of the issue I'm having is that there is disagreement between the initial statement about calibration in the abstract and subsequent text in the manuscript.

We intended our model to match the inverted friction coefficients, but to otherwise remain independent of the field data, so that we can better use it to understand the mechanisms governing the surge generation and propagation that we document. Furthermore, we explicitly did not calibrate the model on the observed depth and velocity data, to avoid circular reasoning when we interpret the results. We have updated the numerical modelling section to make this clearer, as detailed in our response to your excellent comment above.

The second issue I would suggest the authors address head-on is the parameter calibration. There are 8 parameters in this model and it is not clear to me how these parameters were chosen. As these authors know well, using this model or any other model for future prediction is entirely dependent on appropriate parameter choice. If I was a future user of the model, this paper does not give me any clear guidance on how or why I would choose specific parameter values. I think any type of guidance on this would be incredibly valuable in the next version of the manuscript.

As described below, we have updated the manuscript to address this important comment. In general, we didn't intend to suggest that the model can be used for prediction with the current state of knowledge. We hope to revise this position as more data is collected, enabling us to calibrate parameters over multiple flows. However, it may be the case that readers wish to reproduce our procedure with their own data. To address this, we have expanded the Methods and supplementary information section to include more detail on how to choose parameters. This includes various considerations that were made during our parameter selection process that were not explicitly stated and, as a result, has improved the clarity of this section.

The parameters used in our numerical model do not directly correspond to quantities that are measurable in the field, or obtainable from current experimental data. Therefore, Eq. [8] in the main text may be viewed as a phenomenological construction that is flexible enough to approximate the observed (reconstructed) friction values and extend them to a plausible model friction function valid for all $Fr, h > 0$. It is possible to devise many different sets of parameters to those in Table 1 that approximate the friction values constrained by observations. This allows for a wide range of parameter choices that can quantitatively reproduce the essential dynamics of the instability and wave growth discussed in the main text, as shown in Figure S 11.

Nevertheless, there are some useful theoretical considerations that can be used to constrain the selection of parameters, which we detail below. Since the purpose of the simulation in this paper is to validate the reconstructed friction values and establish the relevance of the underlying roll wave instability, we did not perform extensive parameter fitting beyond this. Instead, we conducted several simulations (of which the Figure S 11 cases form a subset) to verify the robustness of our

conclusions to variations in the parameters and chose to present the result with the Table 1 values as a representative example.

Figure S 11: Robustness of numerical results to variations in the frictional parameters. Each panel shows contours of friction, simulated Fr versus h at CD 27 (red) and CD 29 (purple), and the steady balances of Eq. [1] for $\theta = 4.5^\circ$ (lower dashed lines) and $\theta = 3.5^\circ$ (upper dashed lines). Panel A is the simulation presented in the main text, using the parameters from Table 1, while panels B and C use the following modified values: (B) $\mu_1 = -0.025, \mu_2 = 0.12, \beta = 0.5, \beta_* = 0.15, \kappa = 0.2$; (C) $\mu_1 = 0.02, \mu_2 = 0.5, \beta = 8, \beta_* = 0.1, \kappa = 0.01, \Gamma = 1$. Note that the three cases lead to friction functions that are relatively close to the observed friction where data is available, but diverge from one another outside this region. The colorbar shows the friction coefficient.

Three parameters in the friction law, μ_3, L, κ control the behaviour of the friction law near the onset of deposition ($Fr < \beta_*$) and do not play a role in the roll wave instability. The value of μ_3 and the length scale L dictate the range of uniform flows that may exist as static deposits at a given slope angle. Specifically, an initially uniform static deposit of depth h can only remain stationary if $\mu_{start}(h) > \tan \theta$, which rearranges to give

$$h < L \left(\frac{\mu_2 - \mu_1}{\tan \theta - \mu_3} - 1 \right).$$

Since we know that static deposits did form at CD 29, this places a constraint on μ_3 and L (given μ_1 and μ_2 , which are dependent on considerations below). However, without additional data, such as the yield strength of the debris, this does not fully determine these parameters. Therefore, we chose some physically plausible values and verified that simulations are qualitatively insensitive to reasonable adjustments of these selections. (For reference, using our chosen values in the inequality above implies that uniform static layers can form for $h < 1.24\text{m}$ at CD 27 and $h < 2.16\text{m}$ at CD 29.)

The variable κ controls the behaviour of the low Fr frictional regime, which governs the onset of flow arrest and is only accessed intermittently by the waves that form deposits at their tails. A higher value of κ causes deposits to form more rapidly whenever they are triggered and therefore influences the shape of erosion-deposition waves. We set it to a value that produced waves with reasonable qualitative resemblance to the observed profiles at CD 29. Roll waves transition into erosion-deposition waves when their tails drop below $Fr < \beta_*$. Higher values of β_* somewhat enhance the phenomenological distinction between the two wave classes by allowing erosion-deposition waves to form more readily. Our chosen value ($\beta_* = 0.2$) is large enough to promote the formation of clear erosion-deposition waves (as observed in the field), but not so large that it interferes with the roll wave instability that occurs at higher Fr .

The remaining four parameters, determine the friction values in the dynamic regime ($Fr > \beta_*$), within which the initiation and development of roll waves are observed in the field data. Roll waves

emerge from a linear instability of steady uniform flow. Under steady uniform conditions, gravitational forcing is exactly balanced by frictional resistance:

$$\mu_b(Fr, h) = \tan \theta \quad [S1]$$

which, for each slope angle θ , defines a curve in (Fr, h) space. When $Fr > \beta_*$ this may be rearranged to give the linear relationship

$$h = \left(\frac{L}{\beta} \cdot \frac{\mu_2 - \tan \theta}{\tan \theta - \mu_1} \right) (Fr + \Gamma), \quad [S2]$$

Since, h and Fr are both positive, $L/\beta > 0$, and $\tan^{-1} \mu_1$ and $\tan^{-1} \mu_2$ are respectively interpreted as minimum and maximum slope angles upon which uniform steady flows can exist. Therefore, $0 < \tan^{-1} \mu_1 < \theta < \tan^{-1} \mu_2$ (although even permitting $\mu_1 < 0$ can produce a reasonable fit to the reconstructed friction, as demonstrated in Figure S 11B).

To restrict these parameters further, we note that in their idealised description as steady, stable travelling wave solutions to Eqs. [4] and [5] (main text), roll waves are mathematically constrained to contain flow with the values (Fr_0, h_0) at a ‘critical point’ somewhere along their length, where Fr_0 and h_0 are the Froude number and depth of the corresponding uniform flow that gave rise to them via linear instability³⁸. In other words, at a given slope angle, all roll waves must pass through the line defined in Eq. [S2] and the location of this intersection is contingent on the upstream flux (which controls Fr_0, h_0). This can be seen in the simulation data of Figure S 11, which are ‘pinned’ to the black dashed lines. The field data in Figure 4 (main text) are clearly separated into quantitatively different regimes, consistent with the different mean slope angles at the two downstream measurement stations. Therefore, we selected μ_1, μ_2, β , and Γ so that the Eq. [4] lines pass through the field data at CD 27 ($\theta = 4.5^\circ$) and CD 29 ($\theta = 3.5^\circ$). Note that given a systematic procedure for deciding on optimal intersection points, wave data at three different slope angles would be required to uniquely determine μ_1, μ_2, β and Γ . Therefore, we choose illustrative values, which lead to good agreement with the inferred field values of friction, within the available data ranges, as demonstrated in Figure 4 (main text).

My remaining comments are in the line comments below.

Line comments:

13-14. I have a personal preference here that has been reinforced by my organization, that I don’t like these types of grandiose superlative statements because they are rarely true.

We have updated the sentences to clarify that we are referring to in-situ debris flow observations, as opposed to laboratory work. However, we are hesitant to remove all superlatives, for fear of burying the lead. These measurements are really quantitatively different than any in-situ measurements that have come before, achieving laboratory scale detail (as you describe in your comment below) for true field scale debris flows. We think this is a significant advance – one which underpins the entire motivation for this study – and we don’t want to under report that.

23. Just a logic check here. This indicates that it was a series of surges and not just the first surge or one surge out of many that was the most destructive. Do the observations support the fact that the destruction was worse specifically because of multiple surges, or is it more accurate to say something like: “The debris flow was destructive and was observed to have multiple surges which may have each contributed to increased levels of destruction...”. If you aren’t able to truly attribute the source of destruction to surges specifically I don’t think you should make it sound like that in the text.

Yes, your first interpretation is correct here.

26. Here is the danger in superlative statements. For example, Rapstine 2020 recorded 3D data at a speed of 29 Hz. But of course that was in an outdoor flume and focused on initiation. Rengers 2021 recorded 1D data at 60 Hz, again at an outdoor flume. I'm not saying these are exactly comparable, but when you start using language like 'highest resolution...ever made' the reality is that to make that claim you have to add qualifying language to make sure that the statement is true. My suggestion is just to avoid the grand statements so you don't have to spend time qualifying it.

Rapstine, T. D., Rengers, F. K., Allstadt, K. E., Iverson, R. M., Smith, J. B., Obryk, M. K., et al. (2020). Reconstructing the velocity and deformation of a rapid landslide using multiview video. *Journal of Geophysical Research: Earth Surface*, 125, e2019JF005348. <https://doi.org/10.1029/2019JF005348>

Rengers, F. K., Rapstine, T. D., Olsen, M., Allstadt, K. E., Iverson, R. M., Leshchinsky, B., ... & Smith, J. B. (2021). Using high sample rate lidar to measure debris-flow velocity and surface geometry. *Environmental & Engineering Geoscience*, 27(1), 113-126.

We completely agree that we shouldn't oversell our results, but do want to ensure that the novelty is clear. We updated this as below:

*This event was documented by a monitoring system which recorded the highest spatial (~2 cm) and temporal (10 Hz) resolution measurements directly recorded for debris-flow surges **in the field**.*

And added the great suggested references to the following sentence:

In the present work we leverage a series of 3D laser (LiDAR) scanners, originally developed for autonomous driving, high-framerate video cameras, and machine-vision algorithms, which can image natural debris flows at over two orders of magnitude more detail than was previously possible³⁰⁻³⁴.

35. I don't think you mean that the wave is propagating through stationary debris. This makes me visualize something like creating a wave by picking up a rug and creating a wave that moves from one end of it to another. But here I think you mean something like waves propagating over or across initially stationary debris, right?

We actually do mean to say this! We have added some videos to the supplementary information which seem to clearly show compression of material ahead of the wave, suggesting it is more than a surface phenomenon:

Furthermore, if (as we have suggested) the waves are fundamentally similar to erosion-deposition waves observed in dry granular flows, then we can draw upon lab experiments that show such waves mobilising stationary material throughout the flow depth. See (esp. Supplemental Movie 3 of

Edwards, A. N., & Gray, J. M. N. T. (2015). Erosion–deposition waves in shallow granular free-surface flows. *Journal of Fluid Mechanics*, 762, 35-67.

To improve the clarity of the revised manuscript, we updated the sentence to

*Remarkably, this sparse documentation has included accounts of waves propagating through **otherwise** stationary debris^{4,15}*

and have altered our description of erosion-deposition waves elsewhere in the text.

64. I haven't looked at the methods section yet, but it's not immediately clear to me how you get the depth-averaged velocity for any portion of the flow other than the first surge. And even if you restrict it to a single surge, if you have a mobile bed I don't know how you would truly get the depth-averaged velocity from lidar if you can't see the base of the mobile bed.

We make an assumption about the ratio of surface to depth averaged velocity, based on relative velocities of features measured from this and other events, evaluation of the jump conditions, and measurements from other sites. We describe this in the methods:

Next we estimated the width- and depth- averaged velocities for each section

$$\bar{u} = \alpha u_s, \quad [2]$$

where α is the surface velocity reduction factor (0.7) estimated based on the relative velocities of woody debris and boulders^{27,28}, and consistent with other studies^{14,38}.

65. It's a little hard for me to visualize what you are saying here and in the caption from figure 2 where you are saying that the material velocity is slower than the wave velocity. Could you create a conceptual cartoon that demonstrates that with a few arrows to help readers. If you don't have room for more figures maybe this can just be another panel in figure 2. In figure 2 and in the text, I understand that the surges are fairly constant in time and space between stations. So I don't think you are saying that the surges are accelerating. Are you trying to say that if you pick the tops an initial wave (w1) and a subsequent wave (w2) the depth-averaged velocity of the material between w1 and w2 is slower than the speed of either w1 or w2?

Yes, this is exactly right. We have uploaded a video showing a wave crest overtaking particles on the flow surface, which we hope more intuitively demonstrates what we are measuring, and referred to it in the text.

Furthermore, the new Figure 5 and surrounding text in the revised manuscript demonstrates the same point from the perspective of the model simulation. We highlight a wave and show that it propagates faster than the front (which travels at the depth-averaged flow velocity).

83. How did you manually map those?

We describe this in the methods, which we have now referenced in the text:

These features were manually mapped by drawing bounding boxes around identified features in multiple frames, and velocities of the features were derived by calculating the distance between the bounding box centers in subsequent frames, and dividing by the timespan between them.

103. Is the 'erosion-deposition' wave here equivalent to the 'sediment capacitor' described by Kean et al. 2013 (ref. 10)

This is a physically distinct mechanism, as it can happen even in the absence of a change of slope, as shown on the new supplementary Figure S 12. We hope that the addition of Figure 5, as well as the associated text, have now made the specific surge mechanism we are interpreting more clear.

106. Can some of this change in velocity be related to dewatering of the flow?

This could be, however we find it unlikely as the distance between the stations is relatively short (~450 m), which with velocities on the order of 1 to 5 m/s mean that material travels between the two stations within a few minutes. Grain size distributions from the fine material indicate that there is significant clay content, which means low hydraulic conductivities of the material. We therefore doubt significant pore pressure dissipation can occur on this timescale.

127. Since Figure 2A is your map, I'm not sure this is what you mean to refer to in this case to support your statement about basal resistance fluctuations being correlated to large boulders. Do you mean Figure 2B?

Thanks for this comment, we realize we wrote this poorly. We have now completely re-written the discussion, and hope that it is more clear now.

174. The word in this sentence that worries me a bit is 'significant'. I believe here you are talking about the changes between CD27 and CD28. In Figure 2C I see a depth that peaks around 2 m and decreases to about 0.75 m, and in 2D it peaks around 2.5 and decreases to about 1. Those seem different. The velocity in 2C peaks around 5 m/s and decreases to 0, but it looks pretty similar to 2D. It's unclear to me that the velocity changes are statistically significant, and even though the depth in 2C certainly looks to be like a different population of values than the values in 2D, I'm not sure how much a difference in ~0.5 m matters. I'm not saying it doesn't matter, I'm just saying it's not clear to me how important that is from the text.

This is a really good point, and we have reworded this section as below. We hope that the more measured tone better represents our measurement results.

The numerical modelling results demonstrate that the formation and coarsening of surge waves can be quantitatively captured using a single-phase 1D depth-averaged model, so long as the friction law and channel slope angles are appropriately selected (Figure 2 and Figure 4). Given the shallow gradients of the Illgraben fan, a change in mean slope angle of just 1° leads to a measurable change in flow depth and velocity (Figure 2 and Figure 4) that is commensurate with the field data, thereby implying that debris-flow dynamics can be sensitive to seemingly gradual topographic variations⁴⁵. Our simulations cannot capture all the details of the flow, in particular the front shape and behaviour at the tail. This is likely due to the fact that various time-varying properties of the liquefied slurry^{18,46-48} are not accounted for. For example, the field measurements at Gazoduc and CD 27 demonstrate that large boulders locally increase the basal resistance. This leads to the presence of relatively greater frictional variations over the length of the bouldery front, and unsteady flow behind the front, consistent with observations made by other researchers^{9,16,17,49}.

312. Can you say just a bit more about what you did to manually track these? For example, did you pick a static point and track how many frames it took for some observed object to move past the point?

Thanks for this comment, we updated the manual mapping description as follows:

*We validated our surface velocity algorithm by comparing PIV derived velocities with those corresponding to features manually mapped in hillshade projections of the LiDAR point clouds. **These features were manually mapped by drawing bounding boxes around identified features in multiple frames, and velocities of the features were derived by calculating the distance between the bounding box centers in subsequent frames, and dividing by the timespan between them.** As*

shown on Figure 2A, the correspondence is quite high, with maximum values corresponding to woody debris, and minimum values to large boulders.

328. I think the most important thing you can note here is if you have a bedrock bed, a shallow mobile sediment bed above bedrock, or a deep mobile sediment bed. If it's the 3rd choice, then I think that calls into question the ability to actually estimate a depth unless you have some good reasons to believe that a pre-flow channel filled with sediment wouldn't be mobile.

Equation 7. Consider using z to represent elevation rather than y .

This is a great observation, and something that we have now clarified in the text. We think that it is the third choice, however the Illgraben has a series of checkdams installed which stabilize the bed. In fact, when we integrate our discharge timeseries, we estimate volumes of $\sim 21,000 \text{ m}^3$ at Gazoduc, and $\sim 22,000 \text{ m}^3$ at CD 27, with the differences likely in the range of uncertainties surrounding this estimate. We therefore think it is a reasonable assumption to use the pre-event topography as the base.

*We derived estimates of section-averaged depth (defined to be the mean flow depth across the wetted cross-section perpendicular to the bulk flow direction) by assuming a base topography in the channel corresponding to the pre-event topography. **This is likely a reasonable estimate at our study site, as the presence of checkdams stabilizes the bed near our measurement locations.***

389-390. I think it'd be helpful to provide guidance to future users on how they could come up with reasonable values here. Also, it seems like inherent in this sentence you must have done some sort of optimization scheme to determine those values. Any information on how users could determine values for these 8 free parameters would be helpful.

To address this and related comments above, the Methods and supplementary information sections has been substantially expanded with more information, including some guidance on selecting these particular parameters. Please see our response to your excellent comment above

396. Please specify if you can always use the bed-slope here and if you would ever need to adjust this to the friction slope of a flow.

This kind of analysis relies on the mean slope angle being approximately constant so that $dh/dt = dh/dx = 0$ (steady uniform flow), in which case the bed slope is always defines the appropriate gradient.

402. Maybe change 'illustrative values' to 'physically-reasonable values based on X' where X is some reason that makes sense. Either a prior study, or some reasonable constraint.

We elided over the selection of these parameters in the original manuscript to save space. This part of the Methods and the new supplementary material (described above) have been added along the lines you suggest.

419. Is there any way to acknowledge the field observations on Figure 5?

We updated the caption to this Figure to read:

Source condition used in the numerical simulations, **based on a scaled version of the data provided in Figure 2.**

References are incomplete. For example, just looking at 1 and 2 you don't list all the authors, you just use et. al.

Thanks for this comment. The use of the et. al. abbreviation in the bibliography is consistent with the requested reference style from the journal. In the instructions to authors for this journal we find:

For papers with more than five authors include only the first author's name followed by 'et al.'

We have also carefully checked the references to make sure that all are present and didn't find any missing. We would be happy to make changes if you have noticed specific errors.

Figure 1 caption. I think there are two important words that are wrong in this sentence: "Surge wave propagating through stationary material at CD 29." I don't think you mean through, I think you mean something like 'over'. Also, I think you are trying to say that the bed material is initially stationary prior to the surge that flows over it, but you are not claiming that the bed material remains stationary after the surge moves over it. So maybe just change to say 'initially stationary' or something like that to avoid confusion. My main point here is that you are showing a surge that moves across or over material that is initially stationary, but the text here doesn't exactly convey that, which could lead to confusion.

We thank you for this comment. We have updated the text to clarify that the bed material is initially stationary, and added the supplementary videos as described above. However, we have kept the word 'through', as this matches our interpretation of the phenomenon we are observing, as well as other observations of erosion-deposition waves.

Also in the Figure 1 caption, I see a lot of inconsistency in how you write the cite names. I see CD27 on the map, (CD) 27 in the caption, (CD 27) in figure C, and CD 27 in the text. Suggest being more consistent with spaces and parens throughout.

Thanks for this comment, we have updated the caption to read:

A) Overview map of the Illgraben, showing the location of the three observation stations. The distance between the three stations, from Gazoduc downwards, is 1,250 m and 450 m. Image: © swisstopo. B) Bouldery front arrival at the most upstream station, referred to herein as 'Gazoduc'. C) Surge wave at checkdam 27. D) Surge wave propagating through initially stationary material at CD 29. E and F: Surge wave shown on image D at two different times in the LiDAR data, with the white dots showing the pre-event scan of the channel bottom, and the orange dots showing the instantaneous flow surface. The frames are 0.7 s apart.

Figures S1, S3, and S5, it seems a little strange that the orientation of flow changes between these figures. I'm sure there is a good reason, but for consistency would it be possible to make them all flow in the same direction, either toward the top or bottom of the page?

Great point, we have updated this to be more consistent.

REVIEWERS' COMMENTS:

Reviewer #1 (Remarks to the Author):

Thanks for taking the time to address the comments that we provided. I think that the discussion is now enriched and the manuscript is a very useful contribution to the literature. I have no further comments.

Thanks very much for your kind words, and thorough review of the manuscript.

Reviewer #2 (Remarks to the Author):

After reading the authors' reply and the revised manuscript, I found that all of my comments were sincerely addressed and that the manuscript has been improved to a publishable level.

I would like to thank the authors for adding Fig. 5, which clearly helps readers understand the nature of the debris flows focused on in this study. I like it very much. However, please confirm the time labels in the figure. The time intervals are not 90 seconds (as described in the added text at L195), and even assuming 90-second intervals, they do not appear to correspond with the time labels in Fig. 2.

Great catch! We fixed this in the Figure and the text.

I also thank the authors for showing me the comparison between observed and simulated flow discharges. This is exactly the kind of figure I had hoped to see. The comparison clearly shows that, while the numerical simulation successfully reproduces the travel time and flow surface patterns (Fig. 2), it has some difficulty reproducing the flow discharge. For several minutes after the debris-flow front, the simulated discharge appears to be less than half of the actual discharge, indicating that the simulated debris flow is stretched longitudinally. Additionally, at CD29, it is interesting to note that the friction coefficient obtained by the numerical simulation is considerably smaller than that estimated from observations (shown in the new Fig. 3). Although I am not requesting an additional figure, I feel that these pieces of information would be effective for readers to understand the study more deeply.

Good point, we added this Figure to the supplementary information.

Thank you for this interesting paper.

Thanks for your very constructive and thorough review.

Reviewer #3 (Remarks to the Author):

I think that the authors have done a nice job addressing my original comments. I don't have any additional major suggestions at this point.

Thanks again for your thorough review and interesting comments.

I saw in their rebuttal that they were going to try to add a video to the supplemental, and I don't see any videos. But I imagine that they did upload that video and that there must just be something preventing me from seeing it. Similarly, I can't see the simulation you mention in this sentence: "The

full simulation continues for approximately 15 minutes after the final snapshot and may be viewed as an animation provided in the Supplementary Material.”

263: I'd change “debris-flow hazard” to “debris-flow hazards”

Figures 5, 6, and 7: Can you save these figures at a higher resolution? They are blurry on my pdf.

We have uploaded all supplementary videos, and they should now be viewable.